

# Estimating contributions from biomass burning and fossil fuel combustion by means of radiocarbon analysis of carbonaceous aerosols: application to the Valley of Chamonix

Lise Bonvalot[1], Thibaut Tuna[1], Yoann Fagault[1], Jean-Luc Jaffrezo[2], Véronique Jacob[2], Florie Chevrier[2,3], Edouard Bard[1]

[1]CEREGE, Aix-Marseille University, CNRS, IRD, Collège de France, Technopôle de l'Arbois, BP 80, 13545 Aix-en-Provence, France
[2]Université Grenoble Alpes / CNRS, LGGE-UMR 5183, F-38402, Saint Martin d'Hère
[3]Université Savoie Mont-Blanc – LCME, F-73376 Le Bourget du Lac Cedex

*Correspondence to*: Lise Bonvalot (bonvalot@cerege.fr); Edouard Bard (bard@cerege.fr)

**Abstract.** Atmospheric particulate matter (PM) affects the climate in various ways and has a negative impact on human health. In populated mountain valleys from Alpine regions, emissions from road traffic contribute to carbonaceous aerosols, but residential wood burning can be another source of PM during the winter.

We determine the contribution of fossil and non-fossil carbon sources by measuring radiocarbon in aerosols using the recently installed AixMICADAS facility. The accelerator mass spectrometer is coupled to an elemental analyzer (EA) by means of a gas interface system directly connected to the gas ion source. This system provides rapid and accurate radiocarbon measurements for small samples (10-100 µgC) with minimal preparation from the aerosol filters. We show how the contamination induced by the EA protocol can be quantified and corrected for. Several standards and synthetic samples are then used to demonstrate the precision and accuracy of aerosol measurements over the full range of expected $^{14}C/^{12}C$ ratios ranging from modern carbon to fossil carbon depleted in $^{14}C$.

Aerosols sampled in Chamonix and Passy (Arve Valley, French Alps) from November 2013 to August 2014 are analyzed for both radiocarbon (124 analyses in total) and levoglucosan, which is commonly used as a specific tracer for biomass burning. $NO_X$ concentration, which is expected to be associated with traffic emissions, is also monitored.

Based on $^{14}C$ measurements, we can show that the relative fraction of non-fossil carbon is significantly higher in winter than in summer. In winter, non-fossil carbon represents about 85 % of total carbon, while in summer this proportion is still 75 % considering all samples. The largest total carbon and levoglucosan concentrations are observed for winter aerosols with values up to 50 and 8 µg m$^{-3}$, respectively. These levels are higher than those observed in many European cities, but are close to those for other polluted Alpine valleys.

The non-fossil carbon concentrations are strongly correlated with the levoglucosan concentrations in winter samples, suggesting that almost all of the non-fossil carbon originates from wood combustion used for heating during winter.

For summer samples, the joint use of $^{14}C$ and levoglucosan measurements leads to a new model to quantify separately the contributions of biomass burning and biogenic emissions in the non-fossil fraction. The comparison of the biogenic fraction





with polyols (a proxy for primary soil biogenic emissions) and with the temperature suggests a major influence of the secondary biogenic aerosols.

Significant correlations are found between the $NO_X$ concentration and the fossil carbon concentration for all seasons and sites, confirming the relation between road traffic emissions and fossil carbon.

Overall this dual approach combining radiocarbon and levoglucosan analyses strengthens the conclusion concerning the impact of biomass burning. Combining these geochemical data both serves to detect and quantify additional carbon sources. The Arve Valley provides a first illustration of this model to aerosols.

## 1 Introduction

Airborne particles, generally known as atmospheric aerosols or particulate matter (PM), are the focus of many environmental
concerns. Indeed, airborne particles affect the climate on a regional (Penner et al., 1998; Chung and Seinfeld, 2002) and global (Ramanathan et al., 2001a, 2001b) scale by modifying clouds properties (Jacobson et al., 2000), by reflecting, scattering and absorbing sunlight. Notably, the black carbon fraction of PM leads to the second largest anomaly of radiative forcing observed since the beginning of the industrial era, close behind anthropogenic $CO_2$ (Bond et al., 2013).

In addition, the harmful impact of PM on human health is well established: exposure to aerosols can cause respiratory and
cardiopulmonary diseases that lead to increased mortality (Jerrett et al., 2005; Pope and Dockery, 2006; Kennedy, 2007; Lelieveld et al., 2015).

Carbonaceous particles constitute a major fraction of PM (Putaud et al., 2004, 2010). Their sources can be both biogenic and anthropogenic, leading to primary particles (i.e. directly emitted) and to secondary organic particles from gaseous precursors such as volatile organic compounds (Pöschl, 2005).
Improving the characterization of the relative contributions of anthropogenic and natural sources to PM is a crucial issue which has obvious scientific and societal implications (Gustafsson et al., 2009). The importance in PM emission due to biomass burning (BB) for domestic heating has been shown for many urban areas (Jordan et al., 2006b; Zotter et al., 2014). The Arve Valley, located in the French Alps, is strongly impacted by pollution events and high PM concentrations. The severity of these events is due to a combination of topography and local meteorology, notably with temperature inversion layers during winter
which trap air masses close to the ground (Herich et al., 2014). Due to very limited exogenous contributions, notably during winter, the typology of aerosol sources remains simple, which constitute an ideal test site for such measurements.

The pollution of the Arve valley has already been investigated with various techniques and results suggest the influence of local sources of carbon, more specifically from biomass burning used for residential heating during winter (Marchand et al., 2004; Aymoz et al., 2007; Herich et al., 2014). Different sources apportionment models (CMB/PMF/aethalometer) have been
used to determine contribution of the biomass burning in a French alpine city (Grenoble) (Favez et al., 2010) but significant discrepancies due to differences in the conceptual hypotheses made for each model are still observed.



Radiocarbon ($^{14}$C) measurement of the carbonaceous PM fraction has been demonstrated as an effective tool for aerosol source apportionment, in particular for distinguishing fossil fuel combustion products from other carbon sources such as biomass burning and biogenic emissions (Jordan et al., 2006b; Szidat et al., 2006; El Haddad et al., 2011; Liu et al., 2013).

$^{14}$C is produced naturally in the upper atmosphere by the interaction of secondary neutrons from cosmic rays with nitrogen
atoms. It is then oxidized into $^{14}$CO$_2$ and well mixed in the atmosphere before being partly taken up by vegetation during photosynthesis. Living organisms such as trees exhibit $^{14}$C/$^{12}$C ratios similar to that of the atmospheric pool on the order of $10^{-12}$.

Biomass fuel is defined as a generic term meaning a source of modern carbon. Several factors cause the atmospheric $^{14}$C/$^{12}$C ratio to vary slightly from year to year, and this has been well documented over the last decades (Levin and Kromer, 2004;
Hua et al., 2013; Levin et al., 2013). As a consequence, the $^{14}$C/$^{12}$C ratio in the biomass will also vary with the year of growth. By contrast, fossil fuels are depleted in $^{14}$C as they are made of sedimentary organic matter, which is much older than the radioactive half-life of $^{14}$C (T$_{1/2}$ = 5730 years). Therefore, by measuring the $^{14}$C in the whole carbonaceous fraction of aerosol samples, it is possible to quantify the fossil (f$_F$) and non-fossil (f$_{NF}$) fractions.

The direct coupling of an elemental analyzer (EA) to an accelerator mass spectrometer (AMS) is a fast and efficient way to
measure the $^{14}$C in small samples and, more particularly, in aerosols (Ruff et al., 2010a; Salazar et al., 2015). In our case, the CO$_2$ produced by combustion in the EA is delivered into the gas ion source of the AMS AixMICADAS (Bard et al., 2015) by means of the gas interface system (GIS) (Wacker et al., 2013). In contrast to conventional off-line solid AMS analyses (i.e. with a graphitization of the sample), this method can handle very small samples (10-100 µgC). In the case of atmospheric PM samples, such a low required mass allows complementary analyses of other parameters on the same filter.

This study describes our protocol of PM sample analysis for $^{14}$C, including the analyses of standards and blanks in order to quantify and correct for possible contamination (Ruff et al., 2010b). As an example of application, we then determine the fractions of fossil and non-fossil carbon in carbonaceous aerosols from the Arve Valley (French Alps), sampled in the cities of Passy and Chamonix, from November 2013 to August 2014. Levoglucosan, which is a biomass burning molecular proxy (Simoneit et al., 1999), is measured in the same samples and is used to provide an independent view of the biomass burning
contribution. NO$_X$ levels are also monitored in parallel, because they are mainly associated with traffic emissions. Polyols are measured as a proxy for primary biogenic aerosol particles.

## 2 Materials and methods

### 2.1 Radiocarbon measurements: method development

#### 2.1.1 EA-GIS-AixMICADAS coupling

AixMICADAS is a compact AMS system dedicated to $^{14}$C measurements in ultra-small samples (Synal et al., 2007; Bard et al., 2015). It operates at around 200 kV with carbon ion stripping in helium gas. The hybrid ion source works with graphite



targets and $CO_2$ gas (Fahrni et al., 2013; Wacker et al., 2013). It is coupled to a versatile gas interface system that ensures stable gas measurements from different sources: a cracker for $CO_2$ in glass ampoules, an automated system to handle carbonate, and an elemental analyzer for combusting organic matter. AixMICADAS and its performances are described elsewhere (Bard et al., 2015).

Atmospheric PM is collected on quartz filters, but only a small punch (between 0.2 and 1.5 $cm^2$, depending of the filter loading) is required for the $^{14}C$ analysis. The small filter punch is wrapped into a metallic boat before being combusted in the elemental analyzer. The sample preparation is carried out in a laminar flow hood to minimize contamination. The boats are made of silver (10x10x20 mm, about 240 mg each) and are baked at 800 °C for 2 hours to eliminate organic contamination. The EA (VarioMicroCube, Elementar) is equipped with a combustion tube filled with tungsten oxide granules (heated at 1050°C) and

a reduction tube filled with copper wires and silver wool (heated at 550°C). A phosphorus pentoxide trap is then used to retain water produced during combustion and the $CO_2$ is transferred into the zeolite trap of the GIS. $CO_2$ is released by heating the trap to 450°C and is then transferred into the injection syringe by gas expansion. The $CO_2$ is quantified before addition of helium to obtain a 5% $CO_2$ mixture, which is finally injected into the ion source of AixMICADAS. An overall uncertainty of 4 % is considered for the carbon measurements. This conservative value is based on the average difference between duplicate

measurements (2x53) of the same $PM_{10}$ filters. This 4% value thus includes the intrinsic uncertainty of the measurement by the GIS, together with the additional uncertainty linked to loading heterogeneities at the surface of the filters and to the difficulty in punching exactly the same surface of the filter. This 4 % uncertainty is propagated to all values related to the carbon mass.

Measured $^{14}C/^{12}C$ ratios are corrected for fractionation based on the analysis of the $^{13}C$ ion beam on an AixMICADAS Faraday

cup. $^{14}C$ data are then expressed as a normalized activity $F^{14}C$ ratio equivalent to Fraction Modern (Reimer et al., 2004).

Blank measurements are performed using $CO_2$ derived from fossil sources (without $^{14}C$). Measurements of $CO_2$ produced from OxA2 (SRM 4990C, National Institute of Standards and Technology) are used to normalize all $^{14}C/^{12}C$ ratios of the measured samples. Both blank and standard $CO_2$ are contained in bottles directly coupled to the GIS and its injection syringe. During 2015, 85 blank gas samples were measured, giving an average $F^{14}C$ of 0.0045 (SD = 0.0019, N = 85, and $\sigma_{er}$ = 0.0002,

$\sigma_{er}$=SD/$N^{1/2}$). This result is equivalent to a radiocarbon age of 43400 ± 360 years. During the same year, we added 46 OxA2 gas samples, considered as unknown samples, which led to an average $F^{14}C$ of 1.3405 (SD = 0.0064, N = 46, and $\sigma_{er}$= 0.0009). These values are compatible with the standard value of 1.3407 ± 0.0005 $F^{14}C$ (Stuiver, 1983).

In aerosol science, the fraction of modern ($f_M$) is widely used. As underlined by Eriksson Stenström et al. (2011), it is not always clear if $f_M$ has been corrected for decay since 1950 as in Currie et al. (1989). To avoid any confusion in our paper, all

measurements will be expressed in $F^{14}C$ as defined by Reimer et al. (2004). $F^{14}C$ is defined as the ratio of the sample activity to the standard (OxA2) activity measured in the same year, with both activities background-corrected and $\delta^{13}C$ normalized (*i.e.* $A_{SN}/A_{ON}$). $F^{14}C$ does not depend on the year of measurement. Conversion between $F^{14}C$ and $f_M$ (corrected for decay since 1950) is carried out following Eq. (1):



$$f_M = F^{14}C \times \exp\left[(1950 - T_m)\big/8267\right] \tag{1}$$

with $T_m$ the year of measurement and 8267 corresponding to the true mean life of radiocarbon expressed in years, i.e. the true half-life 5730 years divided by ln(2). The exponential factor is slightly lower than one, thus $f_M$ is smaller than $F^{14}C$ (currently about 8 ‰). It is worth underlining that the non-fossil fraction $f_{NF}$ and the fossil fraction $f_F$ do not depend on the $^{14}C$ measurement unit. Indeed, $f_{NF}$ and $f_F$ are ratios between the sample measurement and a reference value, as detailed in Eq. (6), for the modern end-member (the fossil end-member staying at zero). As long as $^{14}C$ measurements and end-members values are expressed in the same unit ($F^{14}C$ or $f_M$), $f_{NF}$ and $f_F$ do not vary with the year of measurement and values determined at different times can be compared.

### 2.1.2 Contamination quantification

It is initially assumed that a sample of a carbon mass $M_S$ and a $^{14}C/^{12}C$ ratio $F^{14}C_S$ analyzed with the EA-GIS coupling becomes contaminated with a constant mass of carbon $M_C$ exhibiting a constant $^{14}C/^{12}C$ ratio $F^{14}C_C$. The main source of contamination is likely to come from the silver boat: while the heat treatment can remove the carbon adsorbed on metallic surfaces of the boat, carbon impurities occluded in the silver cannot be removed. Other sources of carbon may potentially originate in the preparation of the sample (filter) or even from EA-GIS coupling.

The ultimate contamination of metallic boats will be considered as constant. This assumption is expressed in the following mass balance equations Eq. (2) and Eq. (3)) where $M_M$ and $F^{14}C_M$ represent the measured mass and the measured isotopic ratio, respectively (Ruff et al., 2010a):

$$F^{14}C_M \times M_M = F^{14}C_S \times M_S + F^{14}C_C \times M_C \tag{2}$$

$$M_M = M_S + M_C \tag{3}$$

In order to determine $M_C$ and $F_C$ and to test the assumption of constant values, blank and standard samples were measured with various masses $M_M$. Phthalic acid (PA) blank ($F^{14}C = 0$) and OxA2 standard were diluted in ultrapure water and various volumes (less than 25 µL) were deposited onto quartz filter (Pall Flex QAT) punch of approximately 1 cm$^2$ which had been prebaked at 500 °C for 2 hours. Spiked filters punches were wrapped in silver boats then loaded into the EA autosampler. Combining Eq. (2) and Eq. (3) leads to Eq. (4) in which the measured values $M_M$ and $F^{14}C_M$ and the known $F^{14}C_S$ values are used to derive $M_C$ and $F_C$ of the contaminating carbon.

$$F^{14}C_M = \frac{(M_M - M_C) \times F^{14}C_S + M_C \times F^{14}C_C}{M_M} \tag{4}$$

OxA2 and PA samples with different carbon mass ($M_M$) were measured and a nonlinear weighted least squares method (weights corresponding to the measured uncertainties on $F^{14}C_M$ values) was applied to determine $F^{14}C_C$ and $M_C$. The results of the contamination model for the blank and the standard are represented in Fig. 1(a); the estimated parameters from the fit are $F^{14}C_C = 0.73 \pm 0.11$ and $M_C = 1.45 \pm 0.26$ µgC (95 % confidence interval). Figures 1(b) and 1(c) depict the same dataset




corrected for the contamination parameters. It can be observed that $F^{14}C_S$ values for PA and Oxa2 are in agreement with the expected values confirming the constant contamination assumption. Contamination studies were also carried out without filter punches (the blank and standard are laid in solid forms in the silver boats), leading to similar contamination parameters. It may thus be deduced that the boats are the primary source of contamination.

**2.1.3 Standard and synthetic aerosol samples**

In order to mimic aerosol samples, two NIST standards were used as end-members and were mixed together to simulate different $^{14}C/^{12}C$ ratios: SRM 2975 Forklift Diesel Soot (78 % carbon) and SRM 1515 Apple Leaves (45% carbon). The first standard typifies fossil fuel combustion products while the second provides an analog of natural biopolymers generally found in PM (Currie and Kessler, 2005). $^{14}C/^{12}C$ ratios were determined by performing precise measurements on large samples of

roughly 1 mgC that were graphitized with the AGE-3 system and analyzed with AixMICADAS using its hybrid ion source in the conventional mode. As expected, SRM 2975 exhibits a very low $^{14}C/^{12}C$ ratio ($F^{14}C = 0.0013$ with SD = 0.0002, N = 5, and $\sigma_{er} = 0.0001$, blank subtracted) whereas SRM 1515 has the $^{14}C/^{12}C$ ratio of the atmosphere at the time of its photosynthesis in 1985 ($F^{14}C = 1.1862$ with SD = 0.0017, N = 5, and $\sigma_{er} = 0.0007$).

Mixtures of the two SRM standards were prepared to obtain different $^{14}C/^{12}C$ ratios. To ensure homogeneity, the standards

were mixed with an agate mortar and pestle. The relative proportion of modern carbon can be defined as follows in Eq. (5):

$$X_{modern\ carbon} = \frac{mC_{SRM1515}}{mC_{SRM1515}+mC_{SRM2975}} = \frac{0.45 \times m_{SRM1515}}{0.45 \times m_{SRM1515}+0.78 \times m_{SRM2975}} \tag{5}$$

Expected $F^{14}C$ values were calculated by using the mass of each SRM and their measured $F^{14}C$ as end-members. The uncertainties were calculated by propagating different sources of errors: the weighing uncertainty on the mass of each standard and the analytical uncertainties of the $^{14}C/^{12}C$ ratio of the pure standards. All mixed samples were graphitized with the AGE-

3 system and measured with AixMICADAS (three measurements for each mixture). The small scatter of the results listed in Table 1 confirms that mixtures were well homogenized and that $^{14}C/^{12}C$ ratio determinations are reproducible. In addition, the good agreement between theoretical and measured values confirms that these mixtures can be used to simulate small aerosol samples.

Following this initial step, the SRM mixtures were loaded onto quartz filters. In order to simulate real aerosol samples, each

powder mixture was suspended in ultrapure water. Different volumes of these suspensions (about 80 ngC mL$^{-1}$) were then deposited onto quartz filters that had been baked previously at 500 °C for 2h. A vacuum filtration system (Millipore) was used to eliminate most of the water and to distribute carbonaceous particles evenly over the filter surface. Loaded filters were dried overnight in a laminar airflow hood and then subsampled with a puncher (d = 11 mm / S = 0.95 cm$^2$) before being loaded into silver boats. Each standard mixture was measured at least four times with different carbon masses, corresponding to different

loadings on independent filters. Mean results shown in Fig. 2 confirm the accuracy of aerosol measurements with the gas ion source over the full range of expected $^{14}C$ activities ($F^{14}C$ between 0.001 and 1.2).





To further test the precision and accuracy of the developed aerosol analytical procedures, we also analyzed two standards prepared from atmospheric particle matter (Table 2).

We acquired NIST SRM 1649b, prepared from the same bulk material as the original SRM 1649 and SRM 1649a (which are no longer available) but sieved to a smaller particle size fraction (63µm). The original bulk material, SRM 1649 was prepared

at NIST from PM collected in 1976-77 in the Washington DC area over a 12-month period and issued in 1982 (Wise and Watters, 2007, 2009).

High precision measurements were performed to determine the $^{14}C/^{12}C$ ratio of NIST SRM 1649b. Samples were converted to graphite with the AGE 3 system. Four solid targets ($\approx 1$ mgC) were measured. On line gas measurements were also investigated using quartz filters loaded with NIST SRM 1649b. In short, SRM 1649b was suspended in ultrapure water (about 80 ngC mL-

$^1$) and deposited onto previously baked quartz filters. Loaded filters were then dried in the clean hood, punched and wrapped into silver boats, ready for use with the EA-GIS coupled to AixMICADAS. The replicates (N = 7) were obtained with carbon mass ranging from 7 to 93 µg.

Our graphite measurements of large samples are in agreement with the values reported in the literature for SRM 1649 and SRM 1649a(Currie et al., 1984; Currie et al., 2002; Szidat et al., 2004; Wise and Watters, 2007; Heal et al., 2011). The $F^{14}C$

value for the on line gas measurements is 0.505, with a SD of 0.028, N = 7, and a $\sigma_{er}$ of 0.010, whereas the determined $F^{14}C$ for the solid measurements is 0.532 with a SD of 0.004, N = 4, and a $\sigma_{er}$ of 0.002.

Two suggestions could be proposed to explain the small difference between solid and gaseous measurements. Some colloidal fraction or some water-soluble compounds may have been lost during sample preparation. If the soluble and insoluble fractions are of different origins, associated with different isotopic compositions, this could bias the $^{14}C/^{12}C$ ratio of the residual material

loaded on the filter. Similarly, the ultrafine fraction (< 0.3 µm) not retained by the filter may have a different isotopic carbon composition, leading to the discrepancy between the solid and gaseous measurements.

Such a problem does not affect our results on mixtures of SRM 2975 and SRM 1515 standards described previously; indeed, these standards are more prone to be isotopically homogeneous because of their more simple composition as they are both originate from one source.

The second reference material is RM 8785, composed of the fraction lower than 2.5 µm (i.e. PM$_{2.5}$) of SRM 1649 which has been re-suspended in air and deposited onto quartz filters by NIST and SRI International (Cavanagh and Watters, 2005; Klouda et al., 2005). Analyses of three punches give an average $F^{14}C$ of 0.387 and SD of 0.08. This value is in agreement which measurements performed by five different laboratories (Szidat et al., 2013), even if it is positioned at the high end of the values

(Fig. 3). Szidat et al. (2013) pointed out that $^{14}C/^{12}C$ results for RM 8785 exhibit a larger scatter than that measured on other PM samples during the same inter-comparison of laboratories. This was probably caused by heterogeneous loading during production of RM 8785 filters by NIST (concentrations ranging from 92 µg to 2855 µg onto 8.55 cm$^2$ (Cavanagh and Watters, 2005$^)$) or to secondary deposition of volatile organic compounds (VOCs) onto the filters.



An additional source of $^{14}C/^{12}C$ scatter may be linked to the heterogeneity of fine particles (<2.5 μm) constituting RM 8785. Indeed, its average $F^{14}C$ value of approximately 0.39 is quite different from the value of approximately 0.5 measured for SRM 1649a, which was sieved at 125 μm only, and which is the raw material used to produce RM 8785. This suggests the possibility of isotopic heterogeneities between different particle sizes.

## 2.2 Samples from the Arve Valley

### 2.2.1 Sampling sites and procedures

The measurements were performed in the framework of the DECOMBIO program (Chevrier et al., 2016). Filters analyzed in our study were collected between November 2013 and August 2014 in Passy and between December 2013 and January 2014 in Chamonix. Both urban stations, maintained by the local Air Monitoring Agency (Air Rhône-Alpes) are located in the Arve Valley, in the French Alps. The collection sites are presented in Fig. 4. Sampling in the city of Passy (12,000 inhabitants) was performed at 588 m asl (above sea level) whereas sampling in Chamonix (9,000 inhabitants) took place at 1038 m asl. Temperatures were monitored hourly at both sites throughout the sampling period. Daily $PM_{10}$ samples were collected on quartz filter, using DA-80 High Volume Sampler (30 $m^3 h^{-1}$). After collection, filters were folded, wrapped in aluminum foils, sealed in polyethylene bags and stored at -20°C.

### 2.2.2 Additional data

Levoglucosan (1,6-anhydro-b-D-glucopyranose) is an anhydro-sugar, emitted by the pyrolysis of cellulose (Simoneit et al., 1999) and is widely used as a biomass burning tracer (Schauer et al., 2001; Jordan et al., 2006a; Caseiro et al., 2009). Here, the levoglucosan is water extracted and then quantified by HPLC-PAD (Waked et al., 2014). The concentrations of several polyols (arabitol, mannitol, sorbitol) are also determined by this analysis. Polyols at high concentration in the atmospheric PM are known to originate from emission from fungis from soils (Yttri et al., 2007; Bauer et al., 2008).

TC (total carbon) concentration is also quantified on the same filters by the determination of the EC (elemental carbon) and OC (organic carbon) using the thermo-optical method EUSAAR2 (Cavalli et al., 2010). TC is equal to the sum of EC and OC. $PM_{10}$ total mass is measured on line by TEOMS-FDMS, taking into account the volatile and non-volatile fractions of the PM. $NO_X$ ($NO+NO_2$) are also measured on line (with the Environnement S.A. AC32M nitrogen oxides analyzer) and are used as proxies for traffic emissions.

### 2.2.3 Radiocarbon analyses

All samples have been analyzed twice to increase the precision of $^{14}C/^{12}C$ and carbon mass data and to check for possible heterogeneity of individual filters. This represents a total of 124 measurements including the sampling blanks (4 field blanks for Chamonix and 12 for Passy). Blank sampling filters are treated as real samples (in the lab and in the field) with the exception





that no actual sampling is carried out: they are used to ensure that no significant contamination occurs during the different steps of the sampling campaign (e.g. during storage or transport).

Punch surface required for radiocarbon analysis (i.e. punch of 1 cm² or 0.4 cm², depending on the carbon loading of the filter) was determined using the total carbon concentration previously determined by the EC/OC thermo-optical analysis at LGGE.

In this study, the carbon quantity is also determined by the GIS before $CO_2$ injection into the ion source.

The mean carbon mass of the sampling blank filters determined by the GIS system is 1.75 µgC (SD = 1.22 µgC, N = 16). This contamination level agrees with the independent blank assessment described in Sect. 2.1.2. For real aerosol samples, the carbon mass and $^{14}C/^{12}C$ ratios are thus corrected in the same way as described previously.

The carbon content data measured at LGGE and CEREGE are compared in Fig. 5, exhibiting a very strong linear correlation
for both sites (treated together). The slope is close to 1, with a very small intercept, suggesting there is no major difference between measurements obtained on different punch subsamples. This also demonstrates that sampling filters are loaded relatively homogenously.

# 3 Results and discussion

## 3.1 Composition of $PM_{10}$

For both sites, summer samples exhibit daily average $PM_{10}$ concentrations up to 21 µg m⁻³ while winter $PM_{10}$ concentrations range from 13 to 133 µg m⁻³. Twelve days in Passy and three in Chamonix exceed 50 µg m⁻³ and correspond to winter smog episodes, above the public information threshold (see Table 3 and Table 4). On average, winter samples are composed of about 45% carbon (for both Passy and Chamonix), while summer samples in Passy comprise 25% carbon only. The carbon concentration in Passy is very high during the winter season (average of 23 µgC m⁻³), particularly during December with a
mean concentration of 40 µgC m⁻³. It is much lower during July and August at about 3 µgC m⁻³. The mean carbon concentration in Chamonix for December and January is about 18 µgC m⁻³. Therefore, the December average carbon load in Passy is about twice that in Chamonix. Passy is a populated area, located in the lower part of the Arve Valley, with a valley constriction (steep slopes and reduced sun exposure) limiting atmospheric mixing in winter. High emissions and a strong temperature inversion layer persisting for several consecutive days lead to very high particle concentrations when compared to those in Chamonix.

These winter carbon concentrations are an order of magnitude higher than those determined in Gothenburg (Sweden) during February and March 2005 (3 µgC m⁻³) (Szidat et al., 2009) or in Hachioji (Japan) during the 2003 and 2004 winter seasons (less than 3 µgC m⁻³) (Uchida et al., 2010). Comparable concentrations were observed in Switzerland (Szidat et al., 2007), in Roverodo (about 16 µgC m⁻³, January 2005) and Moleno (about 24 µgC m⁻³, February 2005), places that are also typical Alpine valley sites similar to Passy and Chamonix.

The summer mean level of levoglucosan in Passy is close to 0.03 µg m⁻³ which is comparable to summer background concentrations determined by Puxbaum et al. (2007) for six background stations located on an east-west line from Hungary to the Azores. At our sites, winter levels are about 100 times greater than summer ones: in Chamonix, the average concentration



is about 2.6 µg m$^{-3}$ while in Passy about 3.4 µg m$^{-3}$ (up to 8.5 µg m$^{-3}$). These levels are similar to those found in Launceston (Australia) during winter 2003 (Jordan et al., 2006a) but are generally higher than winter levels measured in various European cities (Herich et al., 2014).

Recent studies report that levoglucosan can be partially degraded by photo-oxidation (Hennigan et al., 2010; Kessler et al., 2010) for summer conditions, suggesting that this proxy is not as stable as previously thought. However, as winter temperatures are low (on average between 0 and -2.5 °C) and photo-oxidation is reduced by meteorological conditions during this season, the levoglucosan level is expected to be particularly stable during winter. In addition, in our study, sampling was carried out close to the emissions sources, limiting the exposure time and thus any possible degradation even during summer time. Levoglucosan emission rate depends on various factors, such as the combustion type and conditions (Engling et al., 2006; Schmidl et al., 2008; Lee et al., 2010). Wood type (softwoods and hardwoods) also has an influence on the emission factor of levoglucosan: as ambient measurements generally represents a mixture of different fuels and combustion conditions, the relation between levoglucosan and PM emissions can vary.

### 3.2 $^{14}$C-based source apportionment

Carbon in atmospheric aerosols can originate from both fossil and contemporary sources. Carbon in particles from fossil fuel emissions is characterized by $F^{14}C = 0$, due to the radioactive decay (half-life of 5730 years), whereas $F^{14}C \approx 1$ for carbon in particles coming from contemporary sources. In addition, the atmospheric thermonuclear bomb tests of the late 1950s and early 1960s increased the $^{14}$C content of the atmosphere, leading to $F^{14}C$ contemporary values greater than 1. In the northern hemisphere, the bomb spike reached $F^{14}C$ values on the order of 1.8 in the early 1960s and it has decayed asymptotically since that time (Levin et al., 2010; Hua et al., 2013; Levin et al. 2013). From these studies, a $F^{14}C_{bio} = 1.04$ can be estimated for the atmospheric value for the years 2013-2014.

#### 3.2.1 Apportionment of the carbon pool with a simple hypothesis

In a first and preliminary approximation, we assume that the carbonaceous fraction is composed of both a fossil fraction, without $^{14}$C and so linked to fossil fuels, and an isotopically homogenous non-fossil fraction. To determine this non-fossil fraction ($f_{NF}$), the measured $F^{14}C$ has to be normalized by a non-fossil reference value ($f_{NF,ref}$, expressed in $F^{14}C$) as described by Eq.(6).

$$f_{NF} = \frac{F^{14}C}{f_{NF,ref}} \qquad (6)$$

The high levels of levoglucosan obtained during winter illustrate the significance of biomass burning during this cold season at both sites while summer values suggest that very little biomass burning is recorded for this warm season. Biomass burning is mainly based on wood that grew over the past decades. This means that this carbon fraction integrates an average $F^{14}C$ that is slightly higher than that of the atmosphere at the time of sampling. As per Szidat et al. (2006) and Lewis et al. (2004), we




assume that wood used for biomass burning has an average $F^{14}C_{bb} = 1.10$ ($f_{M,bb} = 1.09$), which can be retrieved from the atmospheric $^{14}C$ record combined with a tree growth model.

For the summer season, it is considered that all non-fossil carbon originates from organic compounds naturally released by living plants (Guenther et al., 1995). This is referred to as the biogenic source of aerosols, whose $F^{14}C$ value should be close

to the atmospheric value at the time of sampling ($F^{14}C_{bio} = 1.04$).

Hence, for this first estimation of the non-fossil and fossil fractions, $f_{NF,ref}$ is estimated to be equal to 1.10 $F^{14}C$ for the winter samples and to 1.04 $F^{14}C$ for the summer ones.

The calculated non-fossil fraction ($f_{NF}$) for the winter samples (Fig. 6) exhibit high values, with mean values equal to 0.89 and 0.84 for Passy and Chamonix, respectively. Lower values observed at Passy in summer (mean $f_{NF} = 0.75$) indicate that the

fossil component is more important in relative term to the total carbon content of aerosols, but that an important non-fossil fraction is still largely dominant.

The concentrations of non-fossil carbon ($TC_{NF}$) and fossil carbon ($TC_F$) can be calculated by multiplying the total carbon concentration TC by the non-fossil fraction ($f_{NF}$) and the fossil fraction ($f_F$) respectively.

While the mean TC is about 13 times larger in winter than summer, the fossil carbon concentration $TC_F$ exhibit a smaller

variation between seasons, as expected from similar traffics over the year. Nonetheless, the $TC_F$ winter concentration is still about 3 times the summer one which may be related to the reduced atmospheric dynamics during winter, leading to trapping of particles by the inversion layers.

Schmidl et al. (2008) demonstrated that combustion of five biomass fuel types (spruce, larch, beech, oak and briquettes) at similar burning conditions leads to a wide range of total carbon to levoglucosan ratios from 4.3 to 17.2. In their study, the TC

only originates from wood combustion and can thus be considered as completely non-fossil TC. The mean $TC_{NF}$/levoglucosan ratios equal 6.2 (SD = 0.4, N =28) for Passy and 6.0 (SD = 0.3, N = 13) for Chamonix. These ratios are within the range, but do not correspond to any particular wood type as presented by (Schmidl et al., 2008). However, the $TC_{NF}$/levoglucosan ratios for Chamonix and Passy are in good agreement to those obtained by Zotter et al. (2014) for several Swiss stations in the south Alps with ratios close to 6.2 ± 2.0, with the exception of Chiasso station ($TC_{NF}$/levo ratio about 9.1 ± 2.6).

The $TC_{NF}$ values are plotted against levoglucosan in Fig. 7, and show a linear relation with high correlation coefficients for Chamonix (Pearson's R = 0.989) and Passy (Pearson's R = 0.995) samples. Moreover, the intercepts are not statistically different from zero showing that virtually all $TC_{NF}$ during the winter originates from the burning of biomass and more specifically from wood combustion used for heating.

One interesting point with these excellent correlations is their stability for a large array of samples, which may include samples

with various aging history and thus variable amount of secondary aerosols produced from VOCs emitted during biomass combustion. Nevertheless, the correlations are established between a primary tracer (levoglucosan) and a total carbon quantity that includes both primary and secondary carbonaceous aerosols. Therefore the excellent correlation implies either that the primary particles are dominant (in general for the total emission or because the secondary formation is slow in our conditions)





or that the fraction of secondary particle is constant in relative terms (i.e. the correlation would remain even if secondary particles were dominant).

As a purely hypothetical case, let's assume that secondary organic aerosols (SOA) vary between 25% and 50% by OC mass of the primary organic aerosols (POA), according to VOC conversion kinetics (i.e. 25% in a recent air and 50% in an older one).

The majority of carbonaceous aerosols would still be composed of primary aerosols, ranging from 80% to 67% of the total carbon, for the two end-members. However, because the dynamic range of total emissions is very large, this variability due to aerosol aging is difficult to detect on the $TC_{NF}$ vs levoglucosan diagram (Fig. 7).

Keeping the same educated guess would imply that for a particular levoglucosan concentration value, one could observe a 20% range of $TC_{NF}$ (i.e. 150/125 = 1.2). To illustrate this on Fig. 7, we show two extreme cases assuming only young air (i.e.

SOA = 25 % of POA) or only older air (i.e. SOA = 50 % of POA). For the same value of levoglucosan, the $TC_{NF}$ ratio between the two extremes should be 1.2, which can be approximated by decreasing or increasing the observed slope by about 10% (dotted lines in Fig. 7) around the observed correlation assumed to be centered between the two end-members. Even if the observed correlation in Fig. 7 is strong, it is clear that its scatter is not completely negligible, but is within the variation between the two hypothetical extremes (see for examples the $TC_{NF}$ values corresponding to about 6 µg m$^{-3}$ of levoglucosan). These

observations and speculations would certainly justify specific studies on secondary aerosol formation processes in the atmospheric conditions of the Arve valley.

During summer, domestic heating emissions are presumed to be weak, as confirmed by really low levoglucosan concentrations. Levoglucosan and $TC_{NF}$ concentrations show no correlation for summer samples (represented by blue dots in Fig. 7). As

mentioned above, $TC_{NF}$ still represents 75% of the total carbon on average in summer. It has already been demonstrated that the modern sources of carbon are dominant over the fossil fuel ones in atmospheric PM of many sites, even in a large city like Marseille, France (El Haddad et al., 2011). It is also the case in these more rural environments. These $TC_{NF}$ levels are about four times higher than expected by the regression model determined for the winter samples, if they were due to biomass burning. These results indicate that the main non-fossil sources differ between seasons. For the winter season, $TC_{NF}$ is directly

related to biomass burning, whereas during summer these sources are most probably biogenic emissions.

To attribute the fossil fraction of the carbonaceous particles, the concentrations of fossil carbon ($TC_F$) are plotted against $NO_X$ concentration (Fig. 8), which is considered as a vehicle emission proxy. Linear correlations are highly statistically significant for all three different datasets. All origin intercepts are equivalent or close to zero. For winter data sets, the slope obtained for the French data sets are roughly equivalent to those given by Zotter et al. (2014) for the $EC_F$ vs. $NO_X$ correlations in many

Swiss sites (no correlations were observed with $OC_F$ in this last study). However, in our study, the slopes for Passy and Chamonix are clearly different, with that in Passy being 50% higher. The reason for such a difference is currently unknown, but may be related to the vehicle fleet influencing the two sites: while the site in Chamonix is an urban traffic site with only passenger vehicles, the site in Passy is an urban background site 1 km away from the highway to Italy supporting a large international truck traffic. Also, the impact of some industrial emissions in Passy remains to be investigated.



The slope obtained for the summer samples (only in Passy) is larger than that obtained for winter, which may suggest different vehicular emissions in summer than in winter or an extensive degradation of $NO_X$ during summer. Another hypothesis is that secondary formation of OC from vehicular gaseous emissions may well be greater in summer than in winter.

### 3.2.2 Apportionment with biogenic fraction variations

The calculations above only constitute a first approximation which takes into account a single non-fossil source to define $f_{NF,ref}$. So far, we considered the non-fossil source to be purely biogenic during summer and to originate exclusively from biomass burning only during winter. However, the non-fossil carbon is made up of these two different fractions which differ slightly in their $^{14}C/^{12}C$ ratios and both have to be acknowledged in the definition of $f_{NF,ref}$.

Zhang et al. (2012) assumed a biogenic fraction ($p_{bio}$) constant throughout the year, implying that its origin does not vary with
the season. More recently, Zotter et al. (2014) applied a variability with the seasons: the biogenic fraction is set at 0.4 during summer and 0.2 during winter, since no large contributions from biogenic sources are expected during the cold season. With these two assumptions, the maximum $F^{14}C$ value in the absence of a fossil component is given by the following mass balance equation Eq. (7) (Szidat et al., 2006; Zhang et al., 2012; Zotter et al., 2014):

$$f_{NF\ ref} = p_{bio} \times F^{14}C_{bio} + (1 - p_{bio}) \times F^{14}C_{bb} \tag{7}$$

where $F^{14}C_{bio}$ and $F^{14}C_{bb}$ correspond to the $F^{14}C$ values of the biogenic (1.04) and the biomass burning components (1.10), respectively. Similarly, $p_{bio}$ corresponds to the biogenic fraction in the total non-fossil carbon, whereas the biomass burning fraction is simply ($1-p_{bio}$). Figure 6 shows the time series of the $f_{NF}$ values calculated by using both the simple and sophisticated models for both sites. In all cases, it must be noted that introducing various values of $p_{bio}$ has a minor impact on $f_{NF,ref}$. Indeed, a decrease of $p_{bio}$ from 1 to 0 would change $f_{NF,ref}$ by 6%. In the same way, the regression parameters for the $TC_{NF}$ vs.
levoglucosan correlations are listed in Table 5 for the different values of $f_{NF,ref}$ (*i.e.* with $p_{bio} = 1$, 0.4, 0.2 and 0). It can be seen that the small variation of $f_{NF,ref}$ has a negligible impact on the linear regression parameters. All approaches confirm the dominance of the biomass burning component during winter as illustrated in Fig. 6.

TableTable 5 also provides all parameters for the $TC_F$ vs $NO_X$ linear fits. Again, correlation coefficients are significant for all values of $p_{bio}$ (1, 0.4, 0.2 and 0) confirming that introducing the hypotheses for this second model is not leading to changes in
the source partitioning.

### 3.2.3 Apportionment for summer samples: independent determination of the non-fossil fraction

One inherent problem with the previous model is that it relies on a priori assumptions about the sources of the non-fossil fraction. In addition, it assumes that the biomass burning and biogenic concentrations ($TC_{bb}$ and $TC_{bio}$) are proportional to $TC_{NF}$ which also implies a linear correlation between the two fractions, i.e. $TC_{bb} = [(1-p_{bio})/p_{bio}] \times TC_{bio}$. Indeed, one could well
imagine a variable emission of biomass burning superimposed on a rather constant emission of biogenic particles, or even a more complex situation as the two sources have different and rather independent origins.



In Sect. 3.2.1, the nearly exclusive contribution of biomass burning to the non-fossil fraction during winter has been demonstrated by the strong linear correlation between levoglucosan and $TC_{NF}$ (Fig. 7 and Table 5) and by intercepts nearly equal to zero (i.e. $TC_{NF} \approx TC_{bb}$ during winter). For summer samples, the insert in Fig. 7 shows that the $TC_{bb}$ expected by the linear models is lower than the measured $TC_{NF}$ suggesting another source of non-fossil carbon.

As an alternative model, we tentatively propose that the part of $TC_{NF}$ due to biomass burning ($TC_{bb}$) in a particular sample could be calculated from its levoglucosan concentration by using its linear correlation to $TC_{NF}$ observed in winter (i.e. $TCbb = TC_{NF} - a \times [levoglocosan]$, a being the slope of the linear relationship shown in Fig. 7).

Total carbon TC is composed of both fossil $TC_F$ and non-fossil fraction $TC_{NF}$. The latter can be sub-divided in parts corresponding to the considered sources, i.e. biomass burning and biogenic emissions ($TC_{bb}$ and $TC_{bio}$ respectively)

$$TC = TC_{NF} + TC_F = TC_{bb} + TC_{bio} + TC_F \tag{8}$$

This leads to the following $^{14}C$ mass balance:

$$TC \times F^{14}C_S = TC_{bb} \times F^{14}C_{bb} + TC_{bio} \times F^{14}C_{bio} + TC_F \times F^{14}C_F = TC_{bb} \times F^{14}C_{bb} + TC_{bio} \times F^{14}C_{bio} \tag{9}$$

With $F^{14}C_S$ the measured $^{14}C/^{12}C$ ratio of the sample, $F^{14}C_{bb} = 1.10$, $F^{14}C_{bio} = 1.04$ and $F^{14}C_F = 0$ as previously discussed in Sect. 3.2.1. TC is the total carbon of the sample [µg m$^{-3}$], $TC_{bb}$ the carbon originating from biomass burning, $TC_{bio}$ the carbon
from biogenic emissions and finally $TC_F$ from fossil sources.

It is thus possible to calculate the biogenic fraction and the fossil fraction by combining the $^{14}C$ mass balance in Eq. (10) and the total carbon mass balance in Eq. (11):

$$TC_{bio} = \frac{F^{14}C_S \times TC - TC_{bb} \times F^{14}C_{bb}}{F^{14}C_{bio}} \tag{10}$$

$$TC_F = TC - TC_{bb} - TC_{bio} \tag{11}$$

The results for summer samples are provided in Table 6. It should be stressed that this model relies on the hypothesis that levoglucosan does not suffer from a large differential degradation between summer and winter, which may be valid to a first order as PM sampling have been carried out close to the emissions sources. The contribution of fossil carbon to TC is estimated to be about 25 %, corresponding to very low fossil carbon concentration i.e. 0.80 µgC m$^{-3}$. By contrast, the results point to a major contribution of about 87 % and up to 93 % of biogenic emissions to the non-fossil fraction (i.e. $p_{bio}$ is about 0.9).

The biogenic carbon concentrations ($TC_{bio}$) can be compared to the concentrations of polyols as these sugar-alcohols are known to be tracers for primary biogenic aerosol particles (Yttri et al., 2007). As shown in Fig. 9, there is no simple relationship between $TC_{bio}$ and polyols for the summer samples, indicating that despite its potential to be a large contributor to PM$_{10}$ in some environments (Waked et al, 2014), this source may not be dominant in the modern fraction of carbon in summer in the Arve Valley.

$TC_{bio}$ includes both primary and secondary organic aerosols, which result from the oxidation of biogenic volatile organic carbon compounds (BVOCs). It is known that BVOCs emissions generally follow the temperature (Leaitch et al., 2011).



Indeed, Fig. 10 shows that $TC_{bio}$ increases with the mean temperature during the warmest part of the day (from 10 am to 6 pm) defining a significant linear correlation (Pearson's coefficient of 0.65 with a slope of $0.27 \pm 0.05$ and y-intercept of $-3.41 \pm 1.06$). Given the physiological effect of temperature, it is logical to expect that emissions are negligible at the low temperature, which can be approximated by an exponential law given in Eq. (12) as in Leaitch et al. (2011).

$$TC_{bio} = \alpha \times \exp(\beta \times T) \tag{12}$$

Where T is expressed in degrees Celsius, $\alpha$ is a constant, which could be assimilated to a base capacity and $\beta$ is an empirical constant. By studying the linear correlation between $\ln(TC_{bio})$ and temperature, it is possible to calculate $\alpha = 0.12 \pm 0.02$ and $\beta = 0.16$ (+0.09 /– 0.06), with a Pearson's coefficient of 0.72. The correlation coefficient is thus slightly higher for the exponential law than for the linear model. In any case, the fact that $TC_{bio}$ depends on temperature suggests that this fraction is
mainly composed of secondary organic aerosol.

**Conclusion**

Quantifying the relative contribution of fossil and non-fossil sources of carbonaceous aerosols is important in order to better understand the sources of atmospheric particles and to attribute them to natural and anthropogenic processes. For example,
both biomass burning for domestic heating and road traffic emissions are known to contribute to PM pollution in many urban areas, notably in the Arve Valley (French Alps) which is the focus of the present study.

Radiocarbon ($^{14}$C) analysis is the best way to distinguish fossil fuel combustion products from other carbon sources such as biomass burning and biogenic emissions. We show here that $^{14}$C is efficiently measured in aerosol samples with the AixMICADAS spectrometer by using an elemental analyzer (EA) coupled to its $CO_2$ gas ion source, which can handle small
samples (10-100 μgC). This direct coupling avoids the production of solid graphite targets, the usual bottleneck in traditional radiocarbon measurement by accelerator mass spectrometry. The present work leads to the following conclusions:

_ Contamination of the measurement procedure is mainly linked to the silver boats in which the filter samples are wrapped prior to combustion in the EA. This contamination has been quantified and shown to be fairly constant, which enables rectification of the measurements of aerosol samples.
_ The precision and accuracy of $^{14}$C measurements in aerosols are validated over the full range of expected fossil and non-fossil carbon values by using various standards and synthetic mixtures.

_ Carbon concentrations of aerosols determined by the LGGE and CEREGE in samples from Passy and Chamonix are in excellent agreement and indicate large concentrations up to 50 μgC m$^{-3}$ during winters.

_ Mean winter carbon concentrations are higher than those reported for several urban sites but are in the range of those
measured in other alpine sites.




_ Levoglucosan content is used as a biomass burning proxy, indicating very high levels during winter with values up to 8 µg m$^{-3}$ in Passy, thus higher than those generally observed in several European cities.

_ Based on $^{14}$C measurements, the fractions of non-fossil carbon determined in winter (0.89 for Passy and 0.84 for Chamonix) are higher than the non-fossil fraction obtained for Passy during summer (0.75). However, the non-fossil fraction remains dominant during summer with a fossil contribution of about 25%, probably from road traffic.

_ Non-fossil carbon concentration (TC$_{NF}$) is strongly correlated with levoglucosan concentration for winter samples (Passy and Chamonix). The linear regression intercepts are close to zero suggesting that almost all of the non-fossil carbon originates from biomass burning and more specifically from wood combustion used for heating during the winter.

_ Fossil carbon concentrations exhibit a strong correlation with NO$_X$ concentrations, suggesting that the source of fossil carbon is directly linked to traffic emissions.

_ Summer samples exhibit an important relative contribution of non-fossil sources (75 %). A dual approach based on $^{14}$C and levoglucosan enables the calculation of a first estimation of the biogenic and biomass burning fractions in the non-fossil carbon. The samples from Passy allow to test this new model, suggesting that for this site the biogenic emissions are the most important contributor to the non-fossil fraction during summer.

_ The lack of correlation between polyols (tracers of biogenic activity in soil) and the biogenic fraction suggest that TC$_{bio}$ could be composed of secondary organic aerosols resulting from the oxidation of biogenic VOC which is also suggests by the correlation between TC$_{bio}$ and temperature.

Overall, combining radiocarbon and levoglucosan measurements strengthens findings concerning the dominant contribution of winter biomass burning to aerosols in the Arve Valley. In addition to first order agreement, both tracers are complementary: levoglucosan enables to identify the source while $^{14}$C allows to precisely quantify the fossil and non-fossil fractions.

We show that this dual approach may also serve to go further in quantifying these additional carbon sources. Combining $^{14}$C and levoglucosan measurements allows reconstructing other contributions such as biogenic aerosols fluxes. As an example, our new model is applied to summer samples from Passy leading to reasonable evaluations of biogenic particle fluxes. These aerosols are probably linked to oxidation of volatile organic compounds (VOCs) as suggested by a significant correlation of fluxes with temperature.

Following Zhang et al. (2012) ongoing research at CEREGE into $^{14}$C measurements of separated elemental carbon and organic carbon fractions of aerosol should provide more precise source apportionment in the future.

**Acknowledgements**

AixMICADAS was acquired and is operated in the framework of the EQUIPEX project ASTER-CEREGE (PI E. Bard) with additional matching funds from the Collège de France, which also supports the salaries of the authors from CEREGE. We thank Sönke Szidat for advice on reporting $^{14}$C results in aerosols.



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



| X modern carbon | Expected $F^{14}C$ | Error | Measurement (after graphitization) | Standard deviation |
|---|---|---|---|---|
| 0 | 0.0013 | 0.0001 | 0.0013 | 0.0002 (N=5) |
| 0.2 | 0.2379 | 0.0185 | 0.2297 | 0.0002 (N=3) |
| 0.51 | 0.6039 | 0.0131 | 0.5896 | 0.0035 (N=3) |
| 0.8 | 0.9467 | 0.0096 | 0.9411 | 0.0012 (N=3) |
| 1 | 1.1862 | 0.0007 | 1.1862 | 0.0017 (N=5) |

**Table 1: Analyses of mixtures of SRM 2975 and SRM 1515 standards in the form of solid graphite targets of large samples (roughly 1 mg C). "Expected $F^{14}C$" is calculated by using the mass of each SRM and its measured $F^{14}C$ as end-members. Measurements are made with solid target (graphitization, roughly 1mgC).**





| Gaseous source $F^{14}C$ | SD (N = 7) | Solid source $F^{14}C$ | SD (N = 4) | Literature values $F^{14}C$ |
|---|---|---|---|---|
| 0.505 | 0.028 | 0.532 $F^{14}C$ | 0.004 | Solid measurement SRM 1649a: 0.523±0.018 $F^{14}C$ (N = 5) (Szidat et al., 2004) 0.507 – 0.61 $F^{14}C$, depending on the sample preparation (Currie et al., 2002; Wise and Watters, 2007) 0.517 / 0.572 $F^{14}C$ (simple /double combustion) (Heal et al., 2011) Solid measurement SRM 1649: 0.61±0.04 $F^{14}C$ (Currie et al., 1984) |

**Table 2: Analyses of SRM1949b with gaseous and solid (roughly 1 mgC) source. Gaseous measurements are made with punches of loaded quartz filters. Comparison with the literature values for SRM 1649 and SRM 1649a.**



| Date | PM$_{10}$ [µg. m$^{-3}$] | Carbon mass [µg m$^{-3}$] | ± Carbon mass [µg m$^{-3}$] | Levoglucosan [µg m$^{-3}$] | NO$_X$ [µg m$^{-3}$] | F$^{14}$C | ± F$^{14}$C | f$_M$ | ± f$_M$ | f$_{NF}$ | f$_F$ | ± f$_F$ / f$_{NF}$ |
|---|---|---|---|---|---|---|---|---|---|---|---|---|
| | | | | | *Winter f$_{NF,ref}$ = 1.10 F$^{14}$C = 1.09 f$_M$* | | | | | | | |
| 24/11/2013 | 31 | 15.95 | 0.46 | 2.37 | Nd | 0.986 | 0.010 | 0.978 | 0.010 | 0.90 | 0.10 | 0.02 |
| *03/12/2013* | *90* | *48.29* | *1.39* | *6.94* | *Nd* | *1.002* | *0.010* | *0.994* | *0.010* | *0.91* | *0.09* | *0.02* |
| *05/12/2013* | *60* | *24.56* | *0.73* | *3.50* | *Nd* | *0.977* | *0.011* | *0.970* | *0.011* | *0.89* | *0.11* | *0.02* |
| *06/12/2013* | *69* | *37.69* | *1.10* | *5.14* | *Nd* | *1.015* | *0.011* | *1.007* | *0.010* | *0.92* | *0.08* | *0.02* |
| *08/12/2013* | *75* | *38.58* | *1.12* | *6.03* | *Nd* | *1.047* | *0.011* | *1.039* | *0.011* | *0.95* | *0.05* | *0.02* |
| *09/12/2013* | *93* | *45.87* | *1.33* | *6.26* | *Nd* | *0.969* | *0.010* | *0.962* | *0.010* | *0.88* | *0.12* | *0.02* |
| *12/12/2013* | *133* | *53.81* | *1.55* | *7.66* | *177* | *0.987* | *0.010* | *0.979* | *0.010* | *0.90* | *0.10* | *0.02* |
| *13/12/2013* | *133* | *57.14* | *1.65* | *8.48* | *170* | *0.985* | *0.010* | *0.977* | *0.010* | *0.90* | *0.10* | *0.02* |
| *15/12/2013* | *86* | *34.46* | *1.01* | *5.81* | *68* | *1.067* | *0.011* | *1.059* | *0.011* | *0.97* | *0.03* | *0.02* |
| *16/12/2013* | *108* | *42.04* | *1.22* | *6.29* | *145* | *0.964* | *0.010* | *0.956* | *0.010* | *0.88* | *0.12* | *0.02* |
| *18/12/2013* | *82* | *28.78* | *0.85* | *4.60* | *120* | *0.926* | *0.010* | *0.919* | *0.010* | *0.84* | *0.16* | *0.02* |
| *20/12/2013* | *60* | *25.20* | *0.74* | *4.20* | *81* | *1.003* | *0.011* | *0.995* | *0.010* | *0.91* | *0.09* | *0.02* |
| 01/01/2014 | 25 | 7.81 | 0.23 | 1.15 | 25 | 0.978 | 0.011 | 0.971 | 0.010 | 0.89 | 0.11 | 0.02 |
| 22/01/2014 | 41 | 17.35 | 0.50 | 2.63 | 57 | 0.957 | 0.011 | 0.949 | 0.010 | 0.87 | 0.13 | 0.02 |
| 12/02/2014 | 21 | 7.80 | 0.23 | 0.97 | 34 | 0.924 | 0.010 | 0.917 | 0.010 | 0.84 | 0.16 | 0.02 |
| 13/02/2014 | 18 | 6.45 | 0.20 | 0.83 | 23 | 0.980 | 0.012 | 0.972 | 0.011 | 0.89 | 0.11 | 0.02 |
| 15/02/2014 | 13 | 3.76 | 0.12 | 0.47 | 13 | 0.875 | 0.013 | 0.868 | 0.013 | 0.80 | 0.20 | 0.02 |
| 16/02/2014 | 43 | 16.48 | 0.48 | 2.61 | 32 | 1.026 | 0.011 | 1.018 | 0.011 | 0.93 | 0.07 | 0.02 |
| *19/02/2014* | *38* | *17.06* | *0.49* | *2.69* | *Nd* | *0.976* | *0.011* | *0.968* | *0.011* | *0.89* | *0.11* | *0.02* |
| *19/02/2014* | *37* | *17.39* | *0.50* | *2.65* | *Nd* | *0.924* | *0.011* | *0.917* | *0.011* | *0.84* | *0.16* | *0.02* |
| 21/02/2014 | 29 | 12.06 | 0.35 | 1.71 | 35 | 1.046 | 0.011 | 1.038 | 0.011 | 0.95 | 0.05 | 0.02 |
| 22/02/2014 | 22 | 7.08 | 0.21 | 0.93 | 24 | 1.011 | 0.011 | 1.003 | 0.011 | 0.92 | 0.08 | 0.02 |
| 24/02/2014 | 38 | 16.17 | 0.39 | 2.09 | 46 | 0.878 | 0.008 | 0.871 | 0.008 | 0.80 | 0.20 | 0.02 |
| 25/02/2014 | 37 | 9.81 | 0.29 | 1.30 | 40 | 0.921 | 0.011 | 0.914 | 0.011 | 0.84 | 0.16 | 0.02 |
| 27/02/2014 | 33 | 12.37 | 0.36 | 1.66 | 46 | 0.956 | 0.011 | 0.949 | 0.011 | 0.87 | 0.13 | 0.02 |
| 28/02/2014 | 20 | 8.81 | 0.26 | 1.31 | 21 | 1.004 | 0.011 | 0.996 | 0.011 | 0.91 | 0.09 | 0.02 |
| 02/03/2014 | 20 | 7.25 | 0.22 | 1.12 | 15 | 1.033 | 0.011 | 1.025 | 0.011 | 0.94 | 0.06 | 0.02 |
| | | | | | *Summer f$_{NF,ref}$ = 1.04 F$^{14}$C = 1.03 f$_M$* | | | | | | | |
| 28/07/2014 | 15 | 3.45 | 0.11 | 0.03 | 14 | 0.712 | 0.012 | 0.708 | 0.012 | 0.69 | 0.31 | 0.01 |
| 30/07/2014 | 12 | 2.81 | 0.10 | 0.03 | 17 | 0.625 | 0.015 | 0.620 | 0.014 | 0.60 | 0.40 | 0.02 |
| 31/07/2014 | 13 | 3.33 | 0.11 | 0.03 | 11 | 0.686 | 0.013 | 0.680 | 0.013 | 0.66 | 0.34 | 0.01 |
| 02/08/2014 | 17 | 3.13 | 0.11 | 0.03 | 10 | 0.815 | 0.014 | 0.809 | 0.014 | 0.78 | 0.22 | 0.02 |
| 03/08/2014 | 13 | 2.81 | 0.10 | 0.04 | 6 | 0.879 | 0.016 | 0.872 | 0.016 | 0.85 | 0.15 | 0.02 |
| 05/08/2014 | 16 | 3.66 | 0.12 | 0.03 | 11 | 0.815 | 0.013 | 0.809 | 0.013 | 0.78 | 0.22 | 0.01 |
| 06/08/2014 | 21 | 4.74 | 0.15 | 0.04 | 15 | 0.771 | 0.011 | 0.765 | 0.011 | 0.74 | 0.26 | 0.01 |
| 08/08/2014 | 10 | 3.17 | 0.11 | 0.02 | 14 | 0.717 | 0.013 | 0.711 | 0.013 | 0.69 | 0.31 | 0.01 |
| 09/08/2014 | 11 | 3.12 | 0.10 | 0.06 | 11 | 0.823 | 0.014 | 0.816 | 0.014 | 0.79 | 0.21 | 0.02 |
| 11/08/2014 | 13 | 2.50 | 0.09 | 0.06 | 13 | 0.771 | 0.017 | 0.764 | 0.017 | 0.74 | 0.26 | 0.02 |
| 12/08/2014 | 13 | 2.73 | 0.10 | 0.05 | 10 | 0.829 | 0.016 | 0.822 | 0.016 | 0.80 | 0.20 | 0.02 |
| 14/08/2014 | 8 | 1.97 | 0.08 | 0.04 | 9 | 0.794 | 0.021 | 0.788 | 0.020 | 0.76 | 0.24 | 0.02 |
| 15/08/2014 | 7 | 2.32 | 0.09 | 0.13 | 9 | 0.890 | 0.020 | 0.883 | 0.020 | 0.86 | 0.14 | 0.02 |
| 17/08/2014 | 9 | 2.39 | 0.09 | 0.04 | 7 | 0.824 | 0.018 | 0.817 | 0.018 | 0.79 | 0.21 | 0.02 |

**Table 3: Results of analysis of Passy samples. PM$_{10}$ is determined by TEOM-FDMS; days with a PM$_{10}$ concentration higher than 50 µg m$^{-3}$ (winter smog) are reported in bold italics (19/02/2014 is sampled with two filters and the sum is greater than 50 µg m$^{-3}$). Levoglucosan and NO$_X$ concentrations: see text. Carbon concentration is determined using the GIS quantification. Each radiocarbon value (expressed in F$^{14}$C and f$_M$) is based on duplicated measurements: here the weighted mean and its weighted error (2σ) is presented. Fossil and non-fossil fractions (f$_F$ and f$_{NF}$) are determined by the radiocarbon measurements.**





| Date | PM$_{10}$ [µg m$^{-3}$] | Carbon mass [µg m$^{-3}$] | ± Carbon masse [µg m$^{-3}$] | Levoglucosan [µg m$^{-3}$] | NO$_X$ [µg m$^{-3}$] | F$^{14}$C | ± F$^{14}$C | f$_M$ | ± f$_M$ | f$_{NF}$ | f$_F$ | ± f$_F$ / f$_{NF}$ |
|---|---|---|---|---|---|---|---|---|---|---|---|---|
| 05/12/2013 | 44 | 19.75 | 0.57 | 2.66 | 174 | 0.900 | 0.011 | 0.893 | 0.011 | 0.82 | 0.18 | 0.02 |
| 08/12/2013 | 44 | 24.80 | 0.73 | 3.83 | 129 | 1.017 | 0.012 | 1.009 | 0.012 | 0.92 | 0.08 | 0.02 |
| *11/12/2013* | *63* | *29.52* | *0.87* | *3.87* | *250* | *0.872* | *0.010* | *0.865* | *0.010* | *0.79* | *0.21* | *0.02* |
| 14/12/2013 | 40 | 20.97 | 0.60 | 3.23 | 133 | 0.968 | 0.011 | 0.961 | 0.011 | 0.88 | 0.12 | 0.02 |
| *17/12/2013* | *53* | *29.97* | *0.88* | *3.45* | *263* | *0.865* | *0.011* | *0.858* | *0.011* | *0.79* | *0.21* | *0.02* |
| 20/12/2013 | 18 | 7.08 | 0.21 | 0.87 | 74 | 0.841 | 0.011 | 0.834 | 0.011 | 0.76 | 0.24 | 0.02 |
| 23/12/2013 | 39 | 20.09 | 0.60 | 3.02 | 170 | 0.914 | 0.011 | 0.907 | 0.011 | 0.83 | 0.17 | 0.02 |
| 26/12/2013 | 14 | 5.73 | 0.18 | 0.79 | 57 | 0.918 | 0.012 | 0.911 | 0.012 | 0.83 | 0.17 | 0.02 |
| 29/12/2013 | 18 | 8.36 | 0.25 | 1.17 | 67 | 0.943 | 0.013 | 0.935 | 0.013 | 0.86 | 0.14 | 0.02 |
| *01/01/2014* | *61* | *26.29* | *0.77* | *4.21* | *154* | *1.018* | *0.012* | *1.010* | *0.012* | *0.93* | *0.07* | *0.02* |
| 04/01/2014 | 17 | 8.48 | 0.25 | 1.23 | 82 | 0.942 | 0.011 | 0.934 | 0.011 | 0.86 | 0.14 | 0.02 |
| 07/01/2014 | 36 | 19.83 | 0.57 | 2.86 | 161 | 0.897 | 0.011 | 0.890 | 0.011 | 0.82 | 0.18 | 0.02 |

**Table 4: Results of analysis of Chamonix samples. PM$_{10}$ is determined by TEOM-FDMS; days with a PM$_{10}$ concentration higher than 50 µg m$^{-3}$ (winter smog) are reported in bold italic. Levoglucosan and NO$_X$ concentrations: see text. Carbon concentration is determined using the GIS quantification. Each radiocarbon value (expressed in F$^{14}$C and f$_M$) is based on duplicated measurements:**
5  **here the weighted mean and its weighted error (2σ) is presented. Fossil and non-fossil fractions (f$_F$ and f$_{NF}$) are determined by the radiocarbon measurements.**



| | TC$_{NF}$ vs. Levoglucosan | | | | | TC$_F$ vs. NO$_X$ | | | | |
|---|---|---|---|---|---|---|---|---|---|---|
| | a | ±a | b | ±b | Pearson's R | a | ±a | b | ±b | Pearson's R |
| *Passy Summer* $p_{bio} = 1$ | - | - | - | - | - | 0.103 | 0.012 | -0.366 | 0.124 | 0.822 |
| *Passy Summer* $p_{bio} = 0.4$ | - | - | - | - | - | 0.099 | 0.014 | -0.270 | 0.145 | 0.813 |
| *Passy Summer* $p_{bio} = 0.2$ | - | - | - | - | - | 0.100 | 0.014 | -0.259 | 0.145 | 0.809 |
| *Passy Summer* $p_{bio} = 0$ | - | - | - | - | - | 0.107 | 0.013 | -0.284 | 0.132 | 0.806 |
| *Passy Winter* $p_{bio} = 1$ | 6.33 | 0.20 | 0.38 | 0.29 | 0.995 | 0.024 | 0.002 | -0.040 | 0.063 | 0.782 |
| *Passy Winter* $p_{bio} = 0.4$ | 6.11 | 0.20 | 0.38 | 0.30 | 0.995 | 0.028 | 0.004 | 0.127 | 0.134 | 0.914 |
| *Passy Winter* $p_{bio} = 0.2$ | 6.04 | 0.20 | 0.37 | 0.30 | 0.995 | 0.031 | 0.004 | 0.129 | 0.133 | 0.935 |
| *Passy Winter* pbio = 0 | 5.98 | 0.19 | 0.36 | 0.28 | 0.995 | 0.036 | 0.003 | 0.062 | 0.086 | 0.950 |
| *Chamonix Winter* $p_{bio} = 1$ | 6.29 | 0.35 | 0.12 | 0.54 | 0.989 | 0.019 | 0.002 | -0.490 | 0.167 | 0.885 |
| *Chamonix Winter* $p_{bio} = 0.4$ | 6.06 | 0.35 | 0.13 | 0.55 | 0.989 | 0.021 | 0.003 | -0.392 | 0.283 | 0.938 |
| *Chamonix Winter* $p_{bio} = 0.2$ | 6.00 | 0.35 | 0.13 | 0.54 | 0.989 | 0.023 | 0.003 | -0.407 | 0.283 | 0.949 |
| *Chamonix Winter* $p_{bio} = 0$ | 5.94 | 0.33 | 0.12 | 0.51 | 0.989 | 0.024 | 0.002 | -0.467 | 0.213 | 0.959 |

**Table 5: Determination of the linear fit parameters (with their 95% confidence intervals) for the linear relation between TC$_{NF}$ and levoglucosan, and for the linear relation between TC$_F$ and NO$_X$. Variation in p$_{bio}$ and in TC$_{NF}$ and TC$_F$ does not have a major influence on the regression parameters in the case of "TC$_{NF}$ vs. Levoglucosan" but does in the case of "TC$_F$ vs. NO$_X$" because of the**

5  **small amount of TC$_F$.**




| | $TC_{bb}$ [$\mu g\ m^{-3}$] | $\pm TC_{bb}$ [$\mu g\ m^{-3}$] | $TC_{bio}$ [$\mu g\ m^{-3}$] | $\pm TC_{bio}$ [$\mu g\ m^{-3}$] | $TC_F$ [$\mu g\ m^{-3}$] | $\pm TC_F$ [$\mu g\ m^{-3}$] | $TC_{NF}/TC$ | $TC_F/TC$ | $TC_{bio}/TC_{NF}$ | polyol [$ng\ m^{-3}$] | $\pm$ polyol [$ng\ m^{-3}$] |
|---|---|---|---|---|---|---|---|---|---|---|---|
| 28/07/2014 | 0.18 | 0.02 | 2.18 | 0.09 | 1.09 | 0.15 | 0.68 | 0.32 | 0.92 | 64.56 | 6.46 |
| 30/07/2014 | 0.21 | 0.02 | 1.47 | 0.08 | 1.14 | 0.12 | 0.60 | 0.40 | 0.88 | 79.88 | 7.99 |
| 31/07/2014 | 0.21 | 0.02 | 1.98 | 0.09 | 1.15 | 0.14 | 0.66 | 0.34 | 0.90 | 74.04 | 7.40 |
| 02/08/2014 | 0.18 | 0.02 | 2.26 | 0.10 | 0.69 | 0.14 | 0.78 | 0.22 | 0.92 | 80.91 | 8.09 |
| 03/08/2014 | 0.26 | 0.03 | 2.10 | 0.10 | 0.45 | 0.14 | 0.84 | 0.16 | 0.89 | 80.23 | 8.02 |
| 05/08/2014 | 0.19 | 0.02 | 2.67 | 0.11 | 0.80 | 0.16 | 0.78 | 0.22 | 0.93 | 65.20 | 6.52 |
| 06/08/2014 | 0.25 | 0.03 | 3.25 | 0.13 | 1.24 | 0.20 | 0.74 | 0.26 | 0.93 | 55.61 | 5.56 |
| 08/08/2014 | 0.14 | 0.02 | 2.03 | 0.09 | 1.00 | 0.14 | 0.69 | 0.31 | 0.93 | 85.26 | 8.53 |
| 09/08/2014 | 0.36 | 0.04 | 2.09 | 0.10 | 0.67 | 0.15 | 0.78 | 0.22 | 0.85 | 76.51 | 7.65 |
| 11/08/2014 | 0.38 | 0.04 | 1.45 | 0.09 | 0.67 | 0.13 | 0.73 | 0.27 | 0.79 | 62.80 | 6.28 |
| 12/08/2014 | 0.27 | 0.03 | 1.89 | 0.09 | 0.57 | 0.14 | 0.79 | 0.21 | 0.87 | 94.51 | 9.45 |
| 14/08/2014 | 0.25 | 0.03 | 1.24 | 0.08 | 0.48 | 0.11 | 0.76 | 0.24 | 0.83 | 50.09 | 5.01 |
| 15/08/2014 | 0.76 | 0.08 | 1.19 | 0.12 | 0.38 | 0.17 | 0.84 | 0.16 | 0.61 | 45.14 | 4.51 |
| 17/08/2014 | 0.25 | 0.03 | 1.63 | 0.09 | 0.51 | 0.13 | 0.79 | 0.21 | 0.87 | 38.93 | 3.89 |

**Table 6: Results of summer samples from Passy. $TC_{bb}$ is calculated from levoglucosan concentration. $TC_{bio}$ and $TC_F$ come out from $TC_{bb}$ and the $F^{14}C$ of the sample. $TC_{NF}/TC$ and $TC_F/TC$ are equivalent to $f_{NF}$ and $f_F$ determined directly by $^{14}C$ measurements. The major part of $TC_{NF}$ is composed of $TC_{bio}$.**



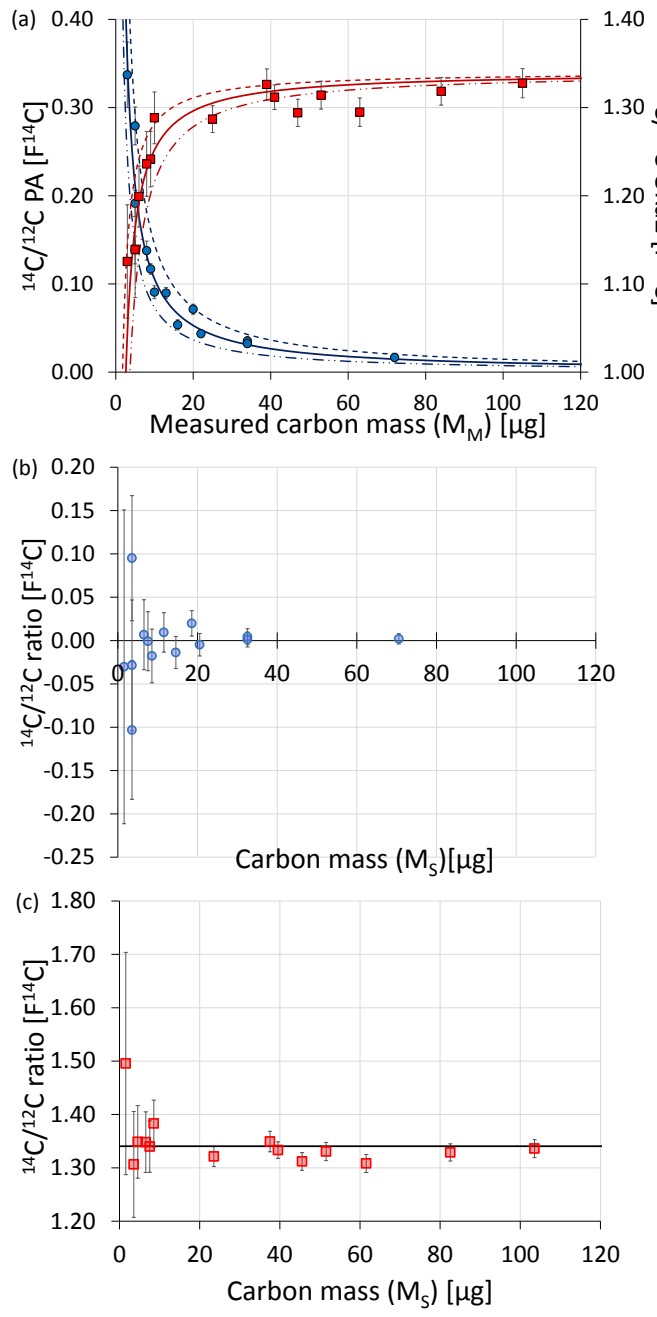

**Figure 1: Measurements and corrections for blank (PA) and standard (Oxa2) samples. (a) Blue dots represents the measured $^{14}C/^{12}C$ ratio for sample blanks and red dots stand for standard measurements. Solid lines and dashed lines represent the least square optimization with its 95% confidence interval. (b) Blue dots are the corrected blank measurements. (c) Red squares are the corrected standard measurements, the line at 1.3406 stands for the certified value of Oxa2. The results exposed in (b) (blank) and (c) (standard) exhibit a good measurement correction.**





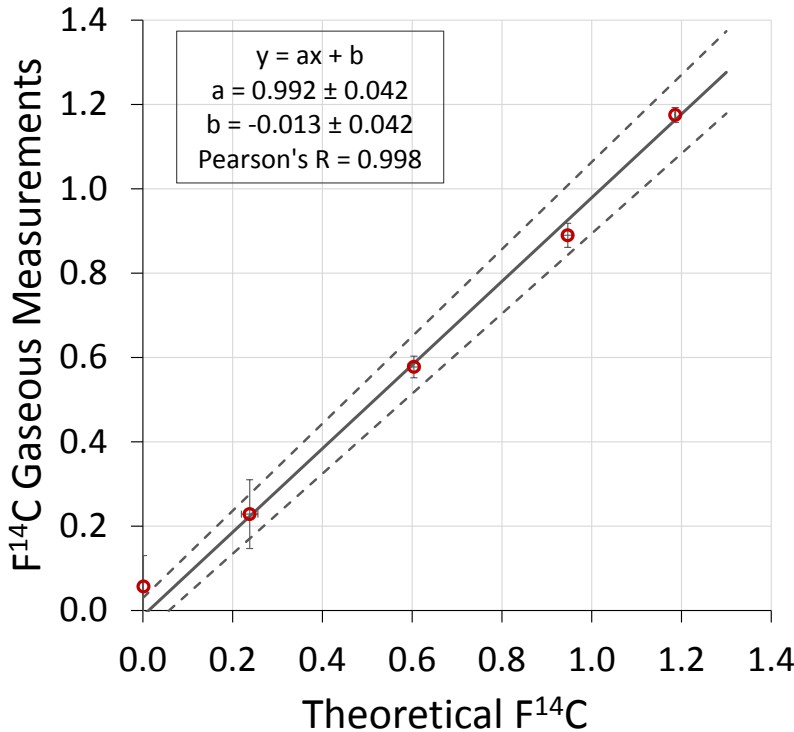

**Figure 2: F$^{14}$C values of simulated aerosol samples measured with the gas source compared with theoretical values. The coefficients of the linear regression have been calculated by taking into account error bars (2 SD) on both axes and are given with their 95 % confidence interval. The linear relation confirms the accuracy of aerosol measurements with the gas ion source over the full range of expected $^{14}$C activities.**





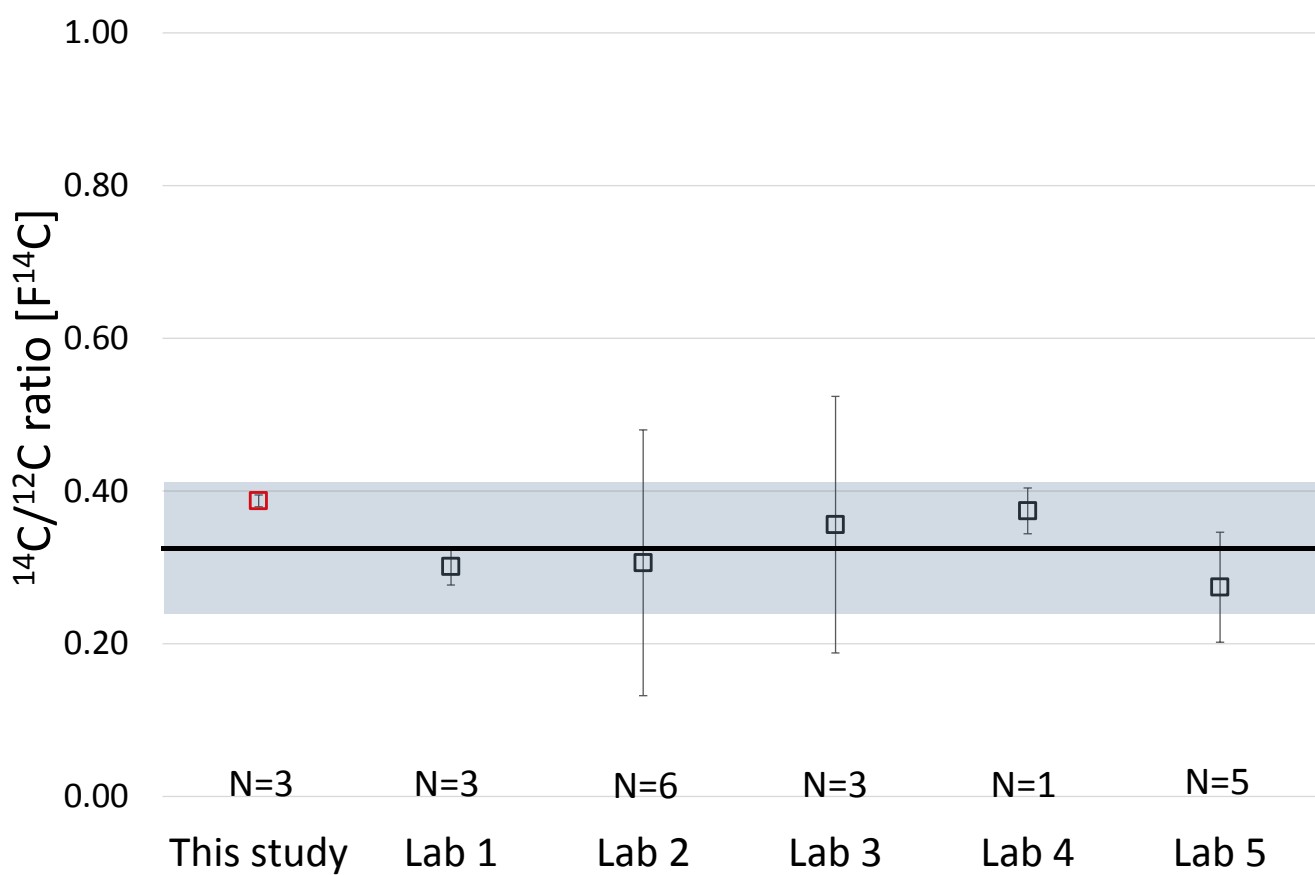

**Figure 3: RM 8785 measurements. Black squares and error bars represent values and measurements uncertainties at 2σ, described in Szidat et al. (2013). The black line stands for the average value and the blue ribbon represents the 2σ confidence interval for Labs 1-5. The red square shows the weighted average result obtained for this study (N = 3) and its weighted error (2σ). A large scatter is exhibit which can be caused by heterogeneous loading during RM 8785 production (Cavanagh and Watters, 2005). The value obtained in this study is compatible with the high end of measurements performed by the five different laboratories.**





**Figure 4: Location of the sampling stations in the Arve valley investigated in this study. PM was sampled between November 2013 and August 2014 in Passy and between December 2013 and January 2014 in Chamonix. Both are urban stations, collecting the PM$_{10}$ fraction of atmospheric aerosols.**





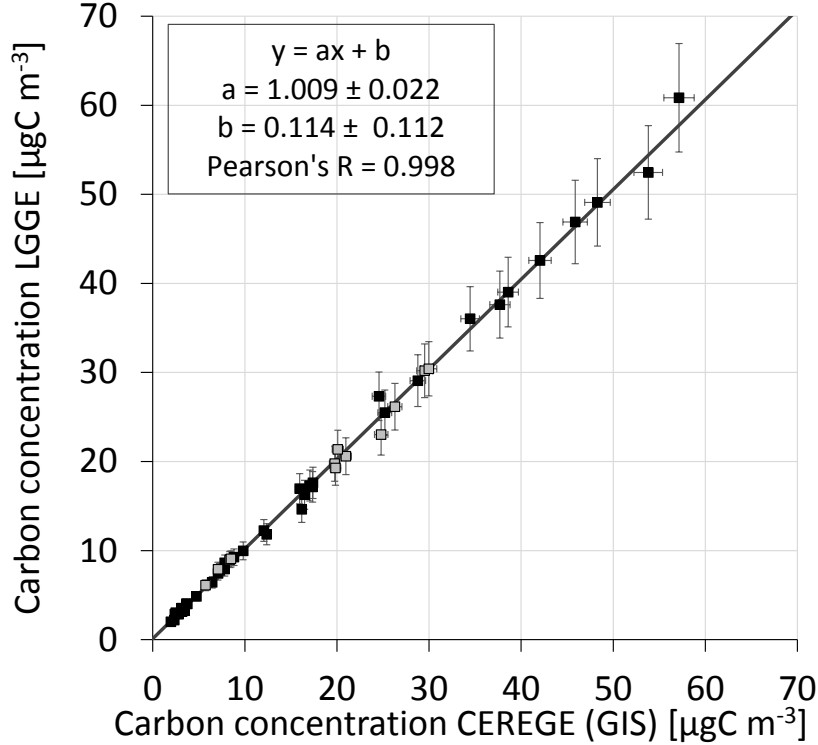

**Figure 5: Total carbon concentration measurements. Comparison between LGGE and CEREGE (GIS) results. The grey squares stand for the Chamonix samples and the black squares stand for the Passy samples. The regression parameters, given with their 95 % confidence intervals, have been calculated by taking into account error bars on both axes and exhibit a very good correlation between the two carbon concentrations; the two measurements can be considered as equivalent.**





**Figure 6: Results for (a) Passy and (b) Chamonix. Grey bars represent carbon concentration. Days with a PM$_{10}$ concentration higher than 50 µg m$^{-3}$ are marked with a yellow star. Green diamonds stand for the only biogenic f$_{NFref}$, pink squares are for f$_{NFref}$ with a 40 % biogenic fraction, purple triangles denote f$_{NFref}$ with a 20% biogenic fraction and red dots stand for f$_{NFref}$ with a 0% biogenic fraction. In both case, the non-fossil fraction remains at very high levels during the winter season, validating the importance of the non-fossil source. A maximum variation of 6 % is observed in the different f$_{NF}$ estimations.**



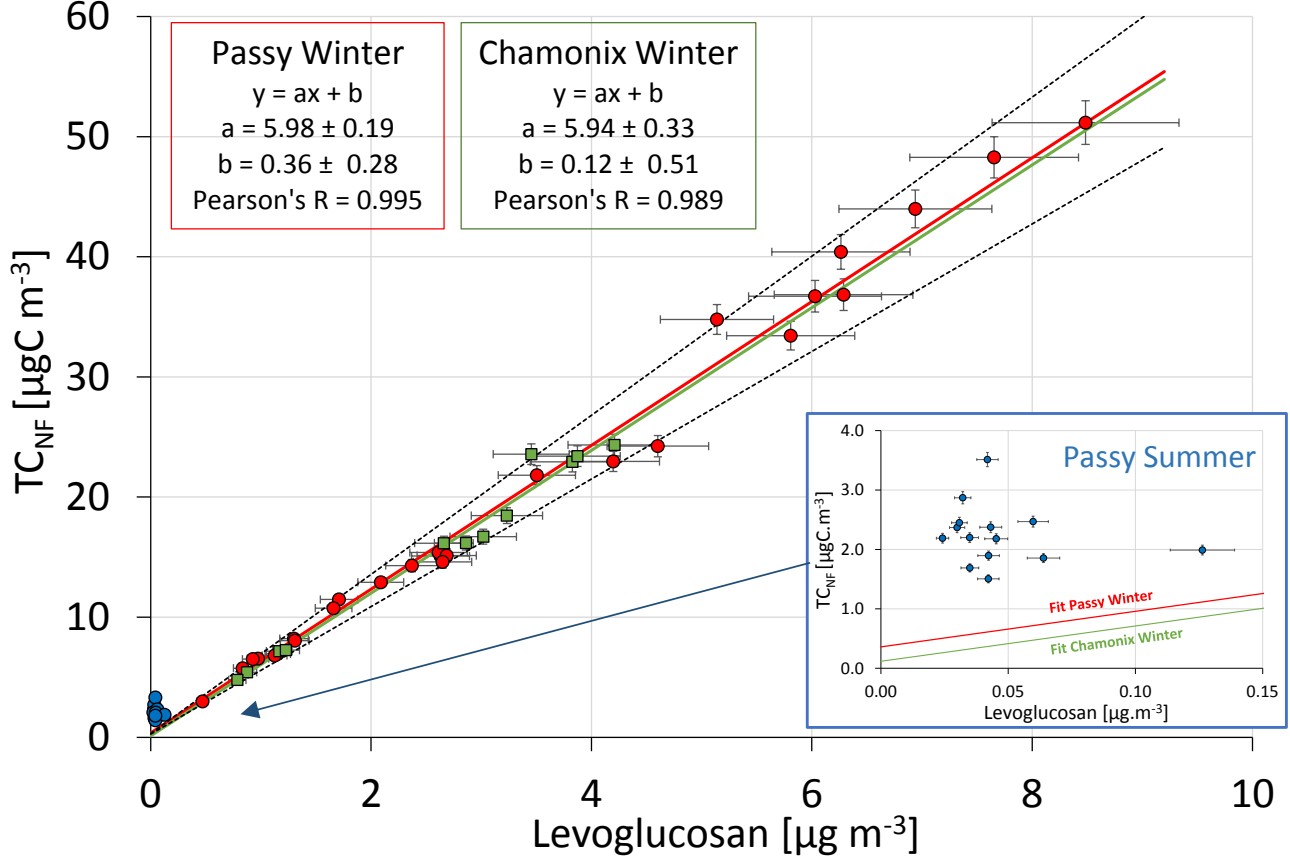

**Figure 7: Comparison of** $Tc_{NF}$**, based on** $^{14}C/^{12}C$ **ratio measurements, with levoglucosan.** $TC_{NF}$ **corresponds to the carbon concentration multiplied by the** $f_{NF}$**. For the winter sample,** $f_{NF}$ **is determined for** $p_{bio} = 0$ **and for summer samples,** $p_{bio} = 1$**. It has to be underlined that a variation in** $p_{bio}$ **does not affect the significance of the relationship between levoglucosan and** $TC_{NF}$ **(see Table**
5 **5). Green squares indicate Chamonix winter samples, red dots Passy winter samples and blue dots denote Passy summer samples. Winter samples display very strong correlations between** $TC_{NF}$ **and levoglucosan with close to zero intercepts suggesting that virtually all of the** $TC_{NF}$ **originates from biomass burning. The fit parameters have been calculated by taking into account both error bars on the x and y axes and are given with their 95 % confidence interval. Black dotted lines stand for two extreme cases assuming only young air (i.e. SOA = 25 % of POA) or only older air (i.e. SOA = 50 % of POA), see Sect. 3.2.1. for further informations. No**
10 **correlation is found for the summer samples, implying the summer** $TC_{NF}$ **originate from other non-fossil sources.**





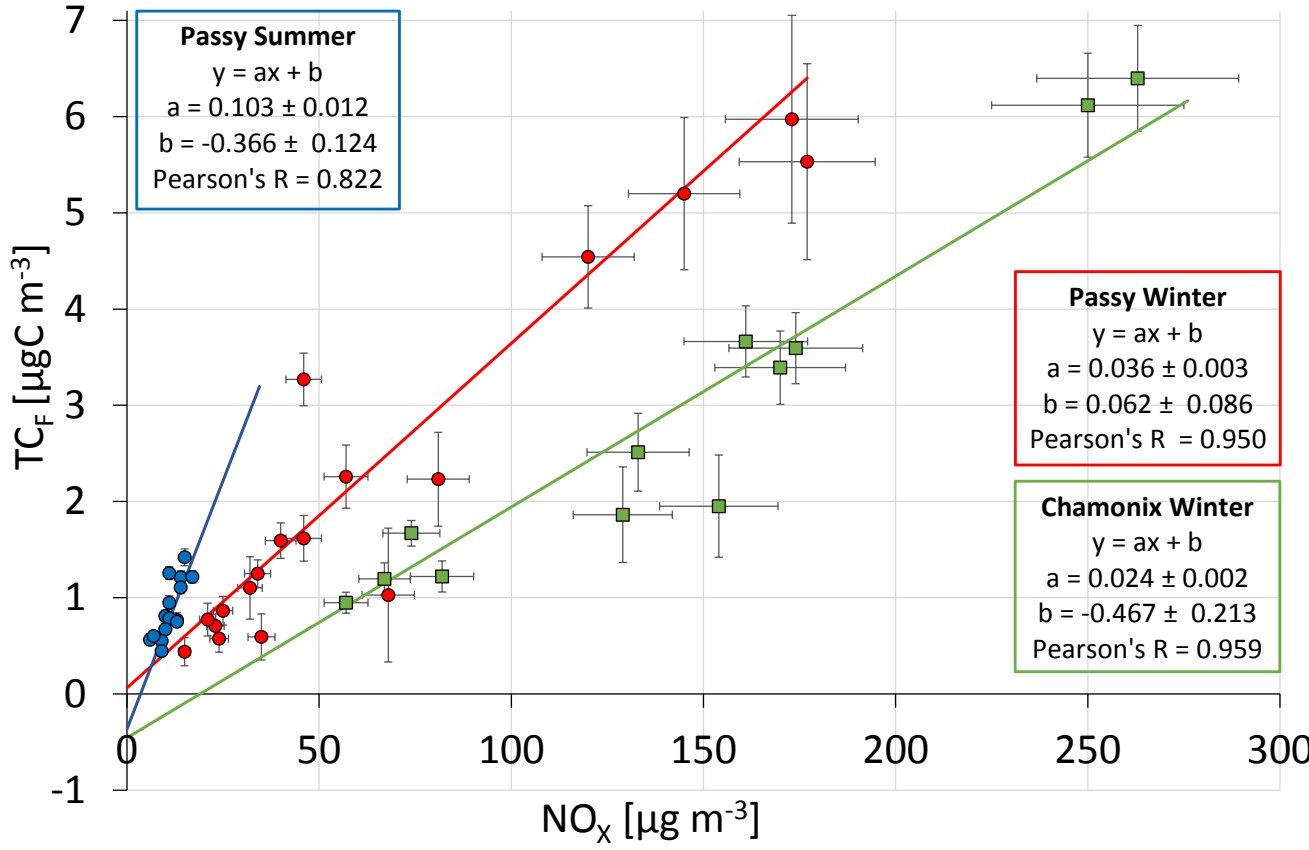

**Figure 8: Comparison of TC$_F$, based on $^{14}$C/$^{12}$C ratio measurements, with NO$_X$. TC$_F$ corresponds to the carbon concentration multiplied by f$_F$. For the winter sample, f$_F$ is determined for p$_{bio}$ = 0 and for summer samples, p$_{bio}$ = 1. It has to be underlined that a variation in p$_{bio}$ does not affect the significance of the relationship between levoglucosan and TC$_{NF}$ (see Table 5). Green squares denote Chamonix winter samples, red dots Passy winter samples and blue dots designate Passy summer samples. Each data set exhibits a good correlation between NO$_X$ and TC$_F$ concentrations. The fit parameters have been calculated by taking into account both error bars on the x and y axes and are given with their 95 % confidence interval. A higher slope value is obtained for the summer data set than for the winter ones, which suggests either different fossil carbon sources or NO$_X$ degradation rate depending on the season.**





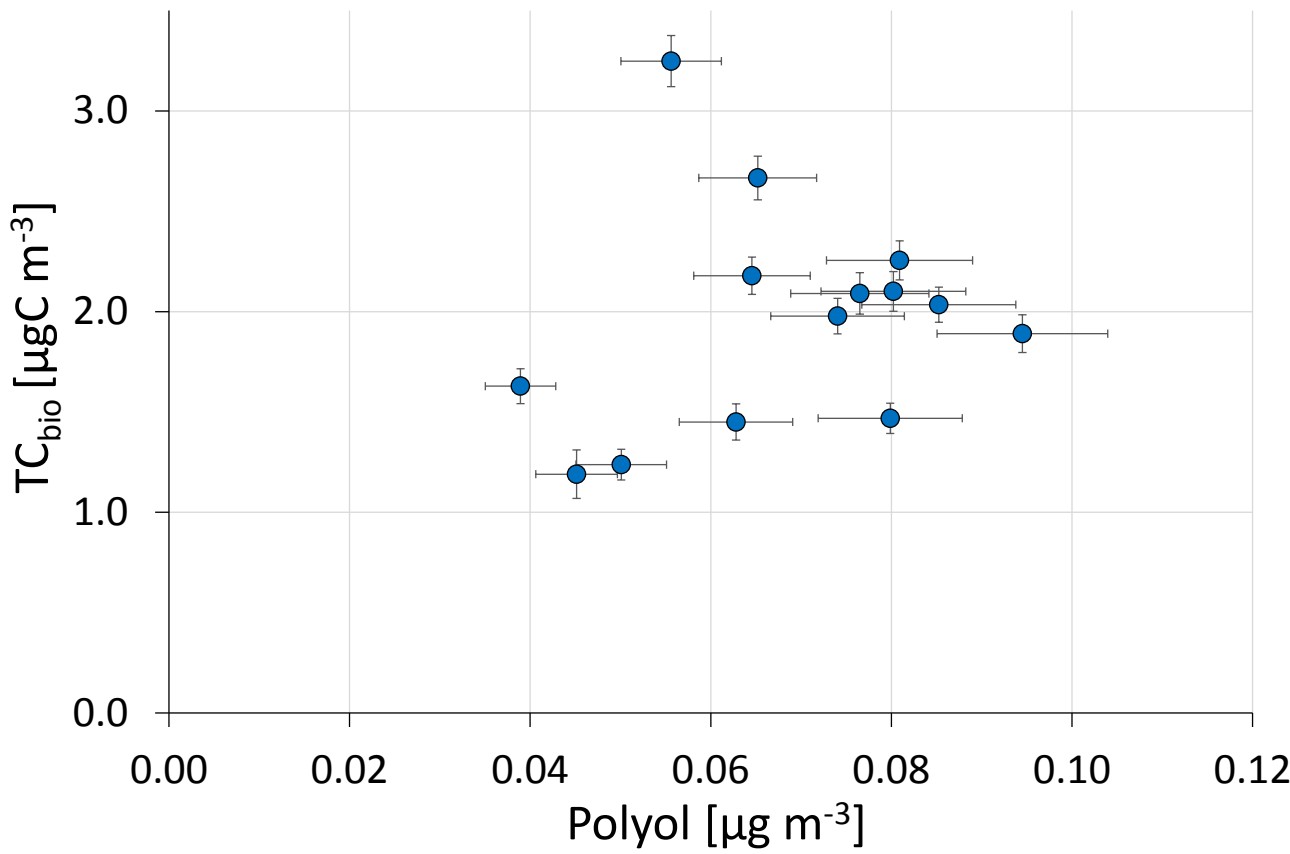

**Figure 9: Comparison of TC$_{bio}$, based on levoglucosan and $^{14}$C/$^{12}$C ratio measurements against polyol concentrations. Blue dots stand for the Passy summer samples. No correlation is found between TC$_{bio}$ and polyols concentrations (primary biogenic emissions tracers). TC$_{bio}$ could originate from secondary organic carbon from the oxidation of biogenic VOC.**





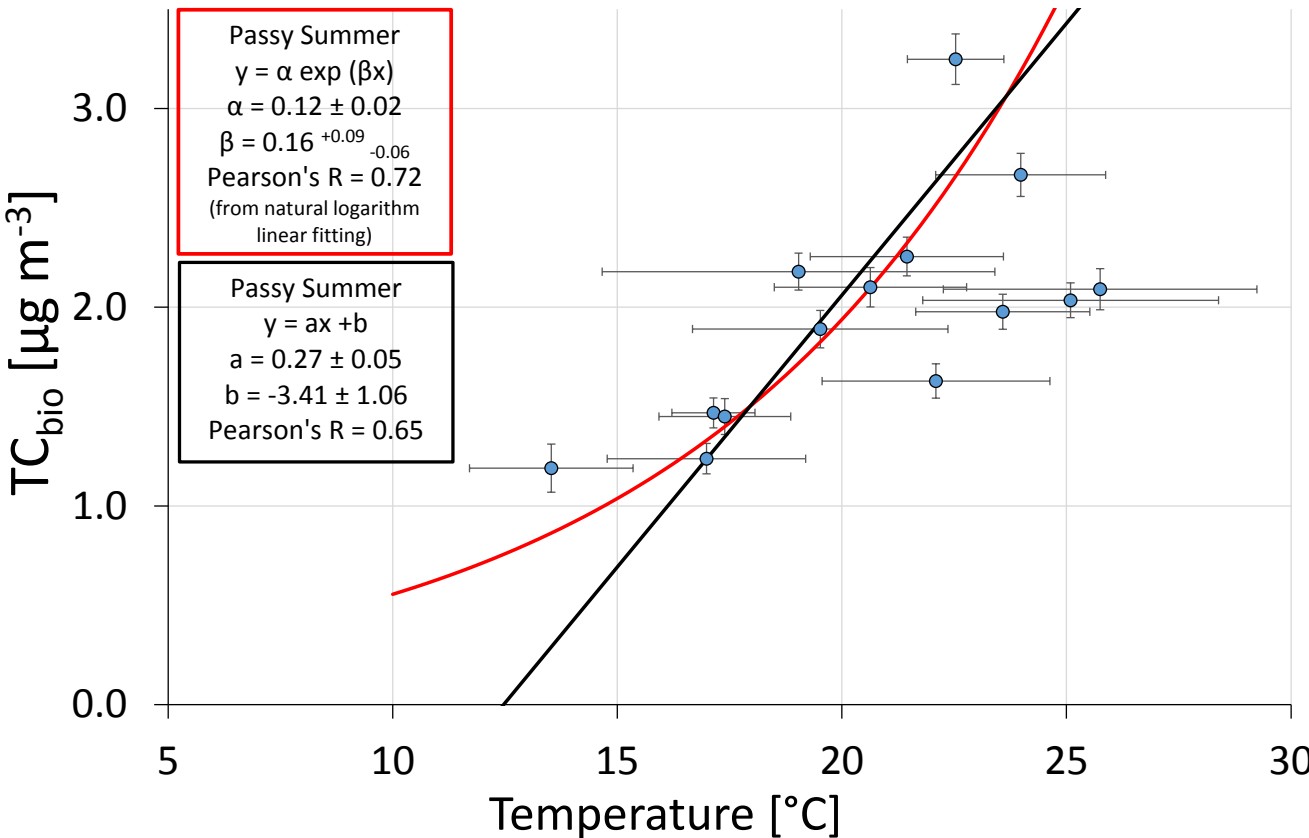

**Figure 10: Comparison of TCbio, based on levoglucosan and 14C/12C ratio measurements plotted versus the average temperature of the warmest part of the day (10 am to 6 pm). Blue dots stand for the Passy summer samples. TCbio concentration increases with the temperature. Both linear (black line) and exponential (red line) relations are represented with their correlation coefficient. The fit parameters have been calculated by taking into account both error bars on the x and y axes and are given with their 95 % confidence interval. The exponential fit is preferred as the TCbio emission cannot be negative. Moreover, emission of BVOCs (precursors of SOA) emission rate is classically described with an exponential law (Leaitch et al., 2011).**