# Peer review of "Estimating contributions from biomass burning, fossil fuel combustion and biogenic carbon to carbonaceous aerosols in the Valley of Chamonix: a dual approach based on radiocarbon and levoglucosan."

_Atmospheric Chemistry and Physics, 2016_

## Referee Comment (RC1) · Anonymous Referee #1 · 5 Jul 2016

**General comments**

The authors present a comprehensive evaluation and validation of the novel EA-GIS-AixMICADAS facility, used to measure radiocarbon without any prior graphitization. The method is also applied to real aerosol samples from the alpine Chamonix Valley. The authors prove the accuracy and precision of the method in a satisfactory manner. Further, great benefit with this facility and method compared to other accelerator mass spectrometers (AMS) is the fact that no graphitization of aerosol samples are needed prior AMS. This makes the method more cost and time efficient. From my experience with graphitization this also means that several errors and sample losses can be avoided.

The applicability to real aerosol samples from the Chamonix Valley show satisfactory results which are in line to what one can expect in terms of source impact during different seasons. The source apportionment model to calculate TC fractions of biogenic, biomass burning and fossil fuel combustion is presented in a clear and concise way and is easily applicable by other researchers for similar studies.

The language is on a clear and high level.

The title is in my opinion to broad and general. It does not say anything about the novel radiocarbon analysis without graphitization. Further, if the authors are about the mention sources in the title as "biomass burning" and "fossil fuel combustion", I am wondering why they don't mention biogenic carbon? This fraction has a considerable role in the results and discussion session in the paper. Finally, the sources were not solely determined by radiocarbon, I would say that levoglucosan was equally important, so why omit levoglucosan?

I recommend this manuscript to be published in ACP.

**Specfic comments**

Page 1, line 15. Please explain the abbreviation AixMICADAS, if not here in the abstract then the first time the abbreviation appears in the following text.

Page 1, line 27. The unit is given in $\mu g.m^{-3}$. Why a dot "." between $\mu g$ and $m^{-3}$, this appears several times in the text but is not consequent. Sometimes this unit is written without a dot between.

Page 2, line 17. Can you please give a number of how large the carbonaceous fraction of PM can be, this would give important knowledge and a feeling for the numbers in this field, especially for new readers.

Page 2, line 21. It feels a bit arrogant to state that there are "obvious scientific and societal implications", I suggest that you erase the word "obvious".

Page 2, line 20-26. Consider re-write this paragraph, it is a bit confusing. End with "ideal test site for such measurements". Which measurements? Those you are referring to in line 20?

Page 3, line 16. The AMS AixMICADAS, state its manufacturer and model name if possible.

Page 3, line 17-18. Other studies have also shown to handle small samples 10-100 µgC with graphitization prior AMS (Genberg et al., 2010).

Page 3, line 31. What do you mean by hybrid ion source? This it can both handle graphite targets and $CO_2$ gas? Please clarify this.

Page 4, line 11. In what atmosphere are you heating the sample to evolve the $CO_2$? This should be stated. Further it is a bit confusing in which temperature the $CO_2$ is evolved, in 450°C or in 1050°C in the EA? Please clarify this.

Page 4, line 15. It is a bit confusing that you mention the sampled $PM_{10}$ filters here prior to the paragraph regarding sampling of filters. Please consider putting the sampling paragraph before the AMS section.

Page 4, line 22. OxA2 is an abbreviation, please spell out the whole name of this standard.

Page 4, line 26. Why did you consider these 46 OxA2 gas samples as unknown samples when you obviously knew the $F^{14}C$ of this SRM?

Page 4, line 33. $A_{SN}/A_{ON}$. Please explain or omit this.

Page 5, line 11-14. I would say that the filter handling and preparation induces most contamination, do you have any reference saying that the silver boat induces large or substantial contamination?

Page 6, line 7. You should say that SRM stands for Standard Reference Material. This is not known to everyone.

Page 6, line 10. What is AGE-3 system? Reference?

Page 8, line 7. DECOMBIO, abbreviation for what?

Page 8, line 10. Please be more specific on the sampling locations. Was it on roofs of buildings? Ground level? How close to the nearest road? Surrounding landscape? Mountains, forests, pastures etc?

Page 8, line 13. Did you prebake the quartz fiber filters to avoid contamination of VOC's prior sampling? What size of filter did you use? What was the brand and model of filters and sampler?

Page 8, line 18. HPLC-PAD, abbreviation for what?

Page 8, line 22. Please state the brand and model of the TOA.

Page 8, line 23. Please state the brand and model of the TEOM-FDMS.

Page 9, line 4. Please explain the abbreviation LGGE.

Page 9, line 9. Please explain the abbreviation CEREGE.

Page 10, line 6. Please mention some meteorological conditions that may reduce photo-oxidation during winter.

Page 10, line 19. Please explain $F^{14}C_{bio}$ to the reader.

Page 11, line 3-4. "For the summer season, it is considered that all non-fossil carbon originates from organic compounds naturally released by living plants". Is this consideration true? What about organic PM from charcoal BBQs? Forest fires?

Page 13, line 15. These $F^{14}C$ values and explanation should have been presented earlier in the manuscript. At page 10 for instance.

Page 14, line 5-7. $TC_{bb}=TC_{NF}-a*[levoglucosan]$. To me, what you propose in this equation is the calculation of $TC_{Bio}$, i.e. $TC_{Bio}=TC_{NF}-a*[levoglucosan]$. $TC_{bb}$ should be: $TC_{bb}=a*[levoglucosan]$, where a is the slope between $TC_{NF}$ and levoglucosan.

With the currently proposed formula, $TC_{bb}$ would be zero (0) during winter which seem highly unrealistic.

Page 15, line 27. Instead of using LGGE and CERGE, which I assume are labs (?), I would prefer if you state the actually used method instead, i.e. EA and TOA. This would make more sense. Either way, you need to explain the abbreviations LGGE and CERGE, which currently are adding confusion to the manuscript.

Page 24, Table 1. "X modern carbon". Please state the unit of this parameter and explain it in the caption of the table.

Page 24, Table 1. Which proportions of each SRM did you use in the mixture?

Page 24, Table 1. "Error". What type of error is this? Should be stated.

Page 24, Table 1. "Measurement after graphitization". Here should be a unit in this column.

Page 25, Table 2. "0.532 $F^{14}C$". Remove "$F^{14}C$" as it is stated in the explaining column text.

Page 26, Table 3. Please explain the "Winter $f_{NF,ref}$=1.10 $F^{14}C$=1.09 $f_M$" that is stated in the table. Same for summer a couple of rows further down. These should be explained in the table caption.

Page 26, Table 3. Column "± Carbon mass $[\mu g.m^{-3}]$". Please state the type of uncertainty, SD, SE, CI?

Page 27, Table 4. Please state the type of uncertainty, SD, SE, CI?

Page 28, Table 5. Please state the type of uncertainty, SD, SE, CI?

Page 29, Table 6. First row, first column. Write "Date".

Page 29, Table 6. Please state the type of uncertainty, SD, SE, CI?

Page 30, Figure 1. Please explain PA, Oxa2, $M_M$ and $M_S$ in the figure caption.

Page 31, Figure 2. Please explain what you mean by "simulated" in the figure caption. From where have you derived the "Theoretical $F^{14}C$", explain. Should further be stated in the figure caption that this graph includes measurements of SRM's.

Page 32, Figure 3. "Blue ribbon", looks green to me. "A large scatter is exhibit which can be caused by…." This sentence sounds erroneous.

Page 34, Figure 5. State that you are comparing EA-GIS and TOA instead of LGGE and CEREGE.

Page 35, Figure 6. Use "TC" or "Carbon Concentration" on the y-axis? Consistency.

**Technical corrections**

Page 5, line 16. Parenthesis error.

Page 13, line 23. Change "TableTable 5" to "Table 5".

Page 14, line 7. Change "[levoglocosan]" to "[levoglucosan]".

Page 25, Table 2. The font is not consistent in the table.

Page 27, Table 4. Change "masse" to "mass".

Whole document: Please be consistent whether you use µgC or just µg. There are discrepancies throughout the whole document, in the tables and figures.

Whole document: Please be consistent whether you use [µg.m$^{-3}$] or [µg m$^{-3}$], same error can be found in ng (nanograms). Personally, I don't see why you use a dot in between. There are discrepancies throughout the whole document, in the tables and figures.

**References**

Genberg, J., Stenstrom, K., Elfman, M., and Olsson, M.: DEVELOPMENT OF GRAPHITIZATION OF mu g-SIZED SAMPLES AT LUND UNIVERSITY, Radiocarbon, 52, 1270-1276, 2010.

---

## Referee Comment (RC2) · Anonymous Referee #3 · 6 Jul 2016

Identification and quantification of the carbon contribution to particulate matter (PM) is crucial for several aspects, such as health, climate and environmental policies. Radiocarbon analysis combined with organic tracers has been demonstrated to be a powerful tool to disentangle modern (e.g. biomass burning) from fossil carbon sources in PM. This paper excels in several aspects from previous work: (1) The newly introduced combination of directly coupled EA to a $CO_2$ gas source of an AMS ion source results in high throughput of very small (10..100 ug) samples, circumventing the costly and time-consuming graphite step. (2) The measurement techniques, including a suite of reference standards and the important assessment of contamination (regarding the

small sample size) are presented in full detail. (3) Due to the exceptionally high sample size and temporal resolution a detailed evaluation of source components of PM in two Alpine valleys is possible and presented convincingly in the paper. I recommend publication in acp without modification.

———————————————

---

## Short Comment (SC1) · 8 Aug 2016

S. Preunkert

preunkert@lgge.obs.ujf-grenoble.fr

This study presents PM10 aerosol data obtained in summer and winter in a valley of the French Alps. Among others, a source apportionment study has been made with the aim to distinguish sources as fossil fuel, biomass burning and biogenic emissions on the base of 14C measurements and levoglucosan. This revealed that summer samples exhibit an important relative contribution of non-fossil sources and a dominant contribution of biomass burning in winter. Interestingly, this very valuable data set and its important conclusions are similar to what was obtained in two source apportionment

studies (Gelencsér et al., 2007; May et al., 2009) made on the basis of a two year round data set sampled on a weekly basis at five rural/remote sites in Europe. These detailed literature data set reflects atmospheric conditions of 2002/2003 on a European west east transect at altitudes from 40 to 3100 m asl. Given the fact that the source apportionment calculations were very similar than ins this study here, i.e. including also 14C and levoglucosan measurements to distinguish fossil, biomass burning and biogenic emissions, it might be worth that the authors have a look on this dataset and benefit by comparing their new results with these existing literature data.

References:

Gelencsér, A., B. May, D. Simpson, A. Sánchez-Ochoa, A. Kasper-Giebl, H. Puxbaum, A. Caseiro, C. Pio, and M. Legrand (2007), Source apportionment of PM2.5 organic aerosol over Europe: Primary/secondary, natural/anthropogenic, and fossil/biogenic origin, J. Geophys. Res., 112, D23S04, doi:10.1029/2006JD008094.

MAY, B., WAGENBACH, D., HAMMER, S., STEIER, P., PUXBAUM, H. and PIO, C. (2009), The anthropogenic influence on carbonaceous aerosol in the European background. Tellus B, 61: 464–472. doi:10.1111/j.1600-0889.2008.00379.x

---

## Author Comment (AC1) · 7 Sep 2016

The authors would like to thank the Reviewer #2 for his/her careful reading and comments, which helped to clarify the manuscript.

The reviewer comments are written in black, our responses in blue and *changes to the manuscript in italic blue.*

**General comments**

The authors present a comprehensive evaluation and validation of the novel EA-GIS-AixMICADAS facility, used to measure radiocarbon without any prior graphitization. The method is also applied to real aerosol samples from the alpine Chamonix Valley. The authors prove the accuracy and precision of the method in a satisfactory manner. Further, great benefit with this facility and method compared to other accelerator mass spectrometers (AMS) is the fact that no graphitization of aerosol samples are needed prior AMS. This makes the method more cost and time efficient. From my experience with graphitization this also means that several errors and sample losses can be avoided.

The applicability to real aerosol samples from the Chamonix Valley show satisfactory results which are in line to what one can expect in terms of source impact during different seasons. The source apportionment model to calculate TC fractions of biogenic, biomass burning and fossil fuel combustion is presented in a clear and concise way and is easily applicable by other researchers for similar studies.

The language is on a clear and high level.

**Specific comments**

The title is in my opinion to broad and general. It does not say anything about the novel radiocarbon analysis without graphitization. Further, if the authors are about the mention sources in the title as "biomass burning" and "fossil fuel combustion", I am wondering why they don't mention biogenic carbon? This fraction has a considerable role in the results and discussion session in the paper. Finally, the sources were not solely determined by radiocarbon, I would say that levoglucosan was equally important, so why omit levoglucosan?

I recommend this manuscript to be published in ACP.

We have complemented the title with both levoglucosan and biogenic carbon.

Page 1, line 15. Please explain the abbreviation AixMICADAS, if not here in the abstract then the first time the abbreviation appears in the following text.

The AixMICADAS abbreviation is explained Page 3 Line 19. The manufacturer name is also stated there.

*AixMICADAS is based on an updated version of the MICADAS (MIni CArbon DAting System) developed and constructed by the ETH Zurich and now produced by the company IonPlus. In contrast to conventional off-line solid AMS analyses where the sample preparation, i. e. graphitization, (Genberg et al., 2010, 2013) of very small samples is complex and time consuming, this method is now applied to very small samples (5-100 µgC) without complex preparation and handling problems.*

Page 1, line 27. The unit is given in µg.m-3. Why a dot "." between µg and m-3, this appears several times in the text but is not consequent. Sometimes this unit is written without a dot between.
All the units have been rewritten without dot.

Page 2, line 17. Can you please give a number of how large the carbonaceous fraction of PM can be, this would give important knowledge and a feeling for the numbers in this field, especially for new readers.
Page 2, line 19. It has been added that at least a third of the PM are composed of carbonaceous compounds.

Page 2, line 21. It feels a bit arrogant to state that there are "obvious scientific and societal implications", I suggest that you erase the word "obvious".
Page 2, line 23. "Obvious" has been removed.

Page 2, line 20-26. Consider re-write this paragraph, it is a bit confusing. End with "ideal test site for such measurements". Which measurements? Those you are referring to in line 20?
Page 2, lines 27-29. The end of this paragraph has been rewritten. Now, it states:
*Due to very limited exogenous contributions, notably during winter, the typology of aerosol sources remains simple, which constitute an ideal site for testing a new method of aerosol sources characterization.*

Page 3, line 16. The AMS AixMICADAS, state its manufacturer and model name if possible.
Page 3 Line 19. The manufacturer has been added. So far, each MICADAS is a prototype with its own improvements (see Bard et al. 2015 *NIM* for AixMICADAS)

Page 3, line 17-18. Other studies have also shown to handle small samples 10-100 µgC with graphitization prior AMS (Genberg et al., 2010).
Page 3, lines 20-23. This reference has been added together with another one from the same group with a focus on aerosol samples (Genberg et al. 2013).
*In contrast to conventional off-line solid AMS analyses where the sample preparation, i. e. graphitization, (Genberg et al., 2010, 2013) of very small samples is complex and time consuming, this method is now applied to very small samples (5-100 µgC) without complex preparation and handling problems.*

Page 3, line 31. What do you mean by hybrid ion source? This it can both handle graphite targets and CO2 gas? Please clarify this.
Page 4, lines 5-6. Indeed, the hybrid ion source can handle graphite target and $CO_2$ gas. The sentence has been rewritten for clarification.
*It is equipped with a hybrid ion source that can both handle graphite targets and $CO_2$ gas*

Page 4, line 11. In what atmosphere are you heating the sample to evolve the CO2? This should be stated. Further it is a bit confusing in which temperature the CO2 is evolved, in 450°C or in 1050°C in the EA? Please clarify this.
Page 4, lines 15-18. The combustion of the sample in the EA and the $CO_2$ transfer process have been detailed.
*The sample is oxidized in the combustion tube under an oxygen-helium atmosphere temporarily enriched with oxygen; the tungsten oxide bed supporting the complete oxidation of combustion gases. Then, the evolved $CO_2$, water and nitrogen oxides flow through the reduction tube (helium is used as carrier gas) where nitrogen oxides are reduced as $N_2$*

Page 4, line 15. It is a bit confusing that you mention the sampled PM10 filters here prior to the paragraph regarding sampling of filters. Please consider putting the sampling paragraph before the AMS section.

Page 4, lines 22-23. We have thus changed the sentence to avoid mentioning the samples. We agree that mentioning the sampled $PM_{10}$ filters of the Arve Valley in this paragraph was confusing, but moving entirely the sampling paragraph before the AMS section would have disrupted the separation between the method protocol (with "home-made" aerosols) and the analysis of the Arve Valley samples.

*This conservative value is based on the average difference between several duplicate measurements of different aerosol samples.*

Page 4, line 22. OxA2 is an abbreviation, please spell out the whole name of this standard.

Page 4, line 30. OxA2 stands for oxalic acid 2 standard. Its definition has been added.

Page 4, line 26. Why did you consider these 46 OxA2 gas samples as unknown samples when you obviously knew the F14C of this SRM?

Page 5, lines 2-4. These additional 46 OxA2 gas samples were considered as samples to measure the average and standard deviation. They are thus completely independent from the other OxA2 measurements, which are used for correction and normalization. This ensures that the mean and standard deviation can be used to assess the accuracy and precision of the AMS measurements (see Bard et al. 2015 NIM).

*OxA2 gas samples are considered as unknown samples so they are not used to correct and normalize measurements (i.e. machine transmission and chemistry fractionation) (Bard et al., 2015) and SD can be quantify.*

Page 4, line 33. ASN/AON. Please explain or omit this.

Page 5, line 9. We added:

*$A_{SN}/A_{ON}$ with $A_{SN}$ the normalized specific activity of the sample and $A_{ON}$ the normalized specific activity of the OxA2.*

Page 5, line 11-14. I would say that the filter handling and preparation induces most contamination, do you have any reference saying that the silver boat induces large or substantial contamination?

Page 5, lines 24-26. The text has been complemented.

*By using the EA, we previously quantified the carbon content of empty silver boats, resulting in a contamination on the order of 1-2 µgC per boat. Similar carbon contaminations have been quantified by Ruff et al. (2010b) for smaller tin boats.*

Page 6, line 7. You should say that SRM stands for Standard Reference Material. This is not known to everyone.

Page 6, line 19. The SRM acronym is explained.

Page 6, line 10. What is AGE-3 system? Reference?

Page 6, lines 22-23. AGE-3 is the graphization system. The acronym is explained and a reference about this system has been added (Wacker, et al. 2010)

Page 8, line 7. DECOMBIO, abbreviation for what?

Page 8, lines 21-23. A definition of the aim of the DECOMBIO project has been added.

*…which focuses on the source apportionment of $PM_{10}$ in the Arve Valley, and the evolution of the contribution of biomass burning emissions (DEconvolution COMBustion BIOmass).*

Page 8, line 10. Please be more specific on the sampling locations. Was it on roofs of buildings? Ground level? How close to the nearest road? Surrounding landscape? Mountains, forests, pastures etc?

Page 8, lines 26-29. Precisions about the sampling locations have been added.

*The collection sites are presented in Fig. 4. Sampling in the city of Passy (12,000 inhabitants) was performed at 583 m asl (above sea level) whereas sampling in Chamonix (9,000 inhabitants) took place at 1035 m asl. For both sampling sites the PM collection occurs about 4 m above the ground. The Passy sampling station is located in a parking lot, 20 m of the closest house and 90 m of a road. The Chamonix sampling occured in the city center, close to shops.*

Page 8, line 13. Did you prebake the quartz fiber filters to avoid contamination of VOC's prior sampling? What size of filter did you use? What was the brand and model of filters and sampler?

Page 8, line 30 – page 9, line 1. Precisions about the filters have been stated.

*Daily $PM_{10}$ samples were collected on quartz filter, using Digitel DA-80 High Volume Sampler (30 $m^3 h^{-1}$). All filters (quartz filters, Pall Tissu Quartz, 150 mm Ø) were pre-baked at 500 °C for 8 h. They were stored in aluminum foil, sealed in a polyethylene sheath before the PM sampling. After collection, filters were folded, wrapped in aluminum foils, sealed in polyethylene bags and stored at -20°C.*

Page 8, line 18. HPLC-PAD, abbreviation for what?

Page 9, lines 6-7. The full technique name is stated.

*… High Performance Liquid Chromatography coupled with Pulsed Amperometric Detection (Dionex, HPLC DX500 and PAD ED40)*

Page 8, line 22. Please state the brand and model of the TOA.

Page 9, lines 11-12. The brand of the TOA is added.

*…by thermal-optical analysis (TOA) EUSAAR2 (Cavalli et al., 2010) with a Sunset apparatus (Birch and Cary, 1996).*

Page 8, line 23. Please state the brand and model of the TEOM-FDMS.

Page 9, lines 12-13. Brand and model of TEOM and FDMS is added.

*(TEOM 1400 ab and FDMS 8500c from Thermo Scientific)*

Page 9, line 4. Please explain the abbreviation LGGE.

Page 9, lines 26 and 28. LGGE and CEREGE are lab names, as listed in the authors affiliations. For clarity, the technique used is stated for each lab.

*The carbon content data measured by TOA in the LGGE (Grenoble) and by the GIS in the CEREGE (Aix-en-Provence)…*

Page 9, line 9. Please explain the abbreviation CEREGE.

Page 9, lines 26 and 28. LGGE and CEREGE are lab names, as listed in the authors affiliations. For clarity, the technique used is stated for each lab.

*The carbon content data measured by TOA in the LGGE (Grenoble) and by the GIS in the CEREGE (Aix-en-Provence)…*

Page 10, line 6. Please mention some meteorological conditions that may reduce photo-oxidation during winter.

Page 10, lines 26-27. We added in parenthesis that the reduction of daylight and strong presence of smog and clouds reduce the photo-oxidation.

Page 10, line 19. Please explain F14Cbio to the reader.

Page 11, lines 7-9. $F^{14}C_{atmo}$ is added for more clarity. $F^{14}C_{atmo}$ is deduced from the literature. As all living systems (like the biomass) exchange with the atmosphere, their radiocarbon level is the same. Therefore $F^{14}C_{atmo} = F^{14}C_{bio}$.

*From these studies, the atmospheric value for the year 2013-2014 can be estimated to $F^{14}C_{atmo} = 1.04$. Hence, biogenic emissions from these years will present the same value ($F^{14}C_{atmo} = F^{14}C_{bio} = 1.04$).*

Page 11, line 3-4. "For the summer season, it is considered that all non-fossil carbon originates from organic compounds naturally released by living plants". Is this consideration true? What about organic PM from charcoal BBQs? Forest fires?

Page 11, lines 23-26. The Arve Valley is not prone to wild fires, and none was recorded during the sampling period. Cholesterol has been quantified in nearly all the samples. This proxy is generally used to quantify the "cooking" influence as it is emitted by cooking meat (charbroiling). The summer samples exhibit very low (often below detection limit) levels of cholesterol, confirming that the influence of meat charbroiling and therefore from BBQ and associated sources can be neglected. The following reference has been added: Schauer, et al. 1999.

*No wild fire was recorded during the sampling period and the influence of the charcoal from barbecue cooking is neglected; levels of cholesterol, generally emitted by meat charbroiling (Schauer et al., 1999) remain very low, pointing that this cooking technique is not important here. Therefore, only the biogenic source of aerosols is considered, whose $F^{14}C$ value should be close to the atmospheric value at the time of sampling ($F^{14}C_{bio} = 1.04$).*

Page 13, line 15. These F14C values and explanation should have been presented earlier in the manuscript. At page 10 for instance.

Page 14, line 7. The $F^{14}C_{bio}$ and $F^{14}C_{bb}$ values are now presented in the 3.2.1 paragraph.

Page 14, line 5-7. TCbb=TCNF-a*[levoglucosan]. To me, what you propose in this equation is the calculation of TCBio, i.e. TCBio=TCNF-a*[levoglucosan]. TCbb should be: TCbb=a*[levoglucosan], where a is the slope between TCNF and levoglucosan.
With the currently proposed formula, TCbb would be zero (0) during winter which seem highly unrealistic.

Page 14, lines 29-30. This was a cut and paste mistake. The equation we used is indeed: $TC_{bb} = a \times [levoglucosan]$

Page 15, line 27. Instead of using LGGE and CERGE, which I assume are labs (?), I would prefer if you state the actually used method instead, i.e. EA and TOA. This would make more sense. Either way, you need to explain the abbreviations LGGE and CERGE, which currently are adding confusion to the manuscript.

Page 16, lines 29-20. Quantification methods have been added. The acronyms are lab names in French as used in the authors affiliations Rather than detailing the long acronyms, we added the town location for each laboratory.

Page 24, Table 1. "X modern carbon". Please state the unit of this parameter and explain it in the caption of the table.

Page 25, Table 1. The X modern carbon state for the mass fraction of modern carbon as defined in Eq. (5), and therefore has no unit. A small definition of the unit has been added in the caption of the table.

Page 25, Table 1. Which proportions of each SRM did you use in the mixture?

Page 25, Table 1. Precision about the determination of X modern carbon has been added, so the reader can determine the SRMs proportions.

*Mass fraction of each SRM can be calculated using their carbon content (i.e. 45% for SRM1 515 and 78 % for SRM 2975.*

Page 24, Table 1. "Error". What type of error is this? Should be stated.
Page 25, Table 1. It is now stated that it is a standard error.

Page 24, Table 1. "Measurement after graphitization". Here should be a unit in this column.
Page 25, Table 1. The unit ($F^{14}C$) has been added.

Page 25, Table 2. "0.532 F14C". Remove "F14C" as it is stated in the explaining column text.
Page 26, Table 2. $F^{14}C$ has been removed.

Page 26, Table 3. Please explain the "Winter fNF,ref=1.10 F14C=1.09 fM" that is stated in the table. Same for summer a couple of rows further down. These should be explained in the table caption.
Page 27, Table 3. These values are now explained in the caption.
*For winter, it is considered that all the non-fossil carbon originates from biomass burning (i.e. $f_{NF,ref} = F^{14}C_{bb}$) whereas all the non-fossil carbon during summer is assumed to originate from biogenic emissions (i.e. $f_{NF,ref} = F^{14}C_{bio}$). The reference values ($f_{NF,ref}$) for winter and summer are expressed in $F^{14}C f_M$. Fossil and non-fossil fractions ($f_F$ and $f_{NF}$) are determined by the radiocarbon measurements (see Eq. (6))*

Page 26, Table 3. Column "± Carbon mass [µg.m-3]". Please state the type of uncertainty, SD, SE, CI?
Page 27, Table 3. The type of uncertainty is now stated (CI).

Page 27, Table 4. Please state the type of uncertainty, SD, SE, CI?
Page 28, Table 4. It is now stated in the caption that uncertainties are confidence intervals.

Page 28, Table 5. Please state the type of uncertainty, SD, SE, CI?
Page 29, Table 5. It is now stated in the caption that uncertainties are confidence intervals.

Page 29, Table 6. First row, first column. Write "Date".
Page 30, Table 6. "Date" has been added.

Page 29, Table 6. Please state the type of uncertainty, SD, SE, CI?
Page 30, Table 6. The type of uncertainty is stated.
*The uncertainties represent the confidence intervals (95 %), and are determined by uncertainties propagation.*

Page 30, Figure 1. Please explain PA, Oxa2, MM and MS in the figure caption.
Page 31, Figure 1. All definitions are stated in the caption.

Page 31, Figure 2. Please explain what you mean by "simulated" in the figure caption. From where have you derived the "Theoretical F14C", explain. Should further be stated in the figure caption that this graph includes measurements of SRM's.
Page 32, Figure 2. The caption is rewritten for clarification. The test aerosols and theirs compositions are defined as well as the theoretical $F^{14}C$.
*$F^{14}C$ values of synthetic and standard (test) aerosol samples measured with the gas source compared with theoretical values. These test aerosols are made of two Standard Reference Materials (SRM 2975 and SRM 1515). The compositions of the different mixtures are listed in Table 1 with the corresponding theoretical and measured $F^{14}C$.*

Page 33, Figure 3. "Blue ribbon", looks green to me. "A large scatter is exhibit which can be caused by…." This sentence sounds erroneous.

Page 34, Figure 3.The color of the ribbon has been changed: a brighter blue is used.
The sentence has been modified:
*The large scatter could be linked to heterogeneous loading during the production of RM 8785 as mentioned by Cavanagh & Watters (2005).*

Page 34, Figure 5. State that you are comparing EA-GIS and TOA instead of LGGE and CEREGE.

Page 35, Figure 5. The figure and its caption have been updated with measurement methods and lab locations.

Page 35, Figure 6. Use "TC" or "Carbon Concentration" on the y-axis? Consistency.

Page 36, Figure 6. The figure has been changed. TC and [µgC m$^{-3}$] are now stated.

**Technical corrections**

Page 5, line 16. Parenthesis error.

Page 5, line 29. The parenthesis is corrected.

Page 13, line 23. Change "TableTable 5" to "Table 5".

Page 14, line 15. The typo is corrected.

Page 14, line 7. Change "[levoglocosan]" to "[levoglucosan]".

Page 14, line 30. The typo is corrected.

Page 25, Table 2. The font is not consistent in the table.

Page 26, Table 2. The font of the table is homogenized.

Page 27, Table 4. Change "masse" to "mass".

Page 28, Table 4. Mass is corrected.

Whole document: Please be consistent whether you use µgC or just µg. There are discrepancies throughout the whole document, in the tables and figures.

µgC is used for carbon mass and µg for other compounds mass.

Whole document: Please be consistent whether you use [µg.m-3] or [µg m-3], same error can be found in ng (nanograms). Personally, I don't see why you use a dot in between. There are discrepancies throughout the whole document, in the tables and figures.

The notation without dot is now used in the whole document.

**References**

Genberg, J., Stenstrom, K., Elfman, M., and Olsson, M.: DEVELOPMENT OF GRAPHITIZATION OF mu g-SIZED SAMPLES AT LUND UNIVERSITY, Radiocarbon, 52, 1270-1276, 2010.

The reference is added to the bibliography.

**Added references**

Birch, M. E. and Cary, R. A.: Elemental Carbon-Based Method for Monitoring Occupational Exposures to Particulate Diesel Exhaust, Aerosol Sci. Technol., 25(3), 221–241, doi:10.1080/02786829608965393, 1996.

Genberg, J., Stenström, K., Elfman, M. and Olsson, M.: Development of graphitization of μ-sized samples at Lund University, Radiocarbon, 52(2–3), 1270–1276, 2010.

Genberg, J., Perron, N., Olsson, M. and Stenstrom, K.: Sealed glass tube combustion of µg-sized aerosol samples, Radiocarbon, 55(2–3), 617–623, 2013.

Schauer, J. J., Kleeman, M. J., Cass, G. R. and Simoneit, B. R. T.: Measurement of Emissions from Air Pollution Sources. 3. C1−C29 Organic Compounds from Meat Charbroiling, Environ. Sci. Technol., 33(10), 1566-1577, doi: 10.1021/es980076j, 1999.

Wacker, L., Němec, M. and Bourquin, J.: A revolutionary graphitisation system: Fully automated, compact and simple, Nucl. Instrum. Methods Phys. Res. Sect. B Beam Interact. Mater. At., 268(7–8), 931–934, doi:10.1016/j.nimb.2009.10.067, 2010.

---

## Author Comment (AC2) · 7 Sep 2016

Identification and quantification of the carbon contribution to particulate matter (PM) is crucial for several aspects, such as health, climate and environmental policies. Radiocarbon analysis combined with organic tracers has been demonstrated to be a powerful tool to disentangle modern (e.g. biomass burning) from fossil carbon sources in PM. This paper excels in several aspects from previous work: (1) The newly introduced combination of directly coupled EA to a CO2 gas source of an AMS ion source results in high throughput of very small (10..100 ug) samples, circumventing the costly and time-consuming graphite step. (2) The measurement techniques, including a suite of reference standards and the important assessment of contamination (regarding the small sample size) are presented in full detail. (3) Due to the exceptionally high sample size and temporal resolution a detailed evaluation of source components of PM in two Alpine valleys is possible and presented convincingly in the paper. I recommend publication in acp without modification.

We thank the Anonymous Referee #3 for his/her comment underlining the strongest points of our paper.

---

## Author Comment (AC4) · 7 Sep 2016

S. Preunkert

preunkert@lgge.obs.ujf-grenoble.fr

This study presents $PM_{10}$ aerosol data obtained in summer and winter in a valley of the French Alps. Among others, a source apportionment study has been made with the aim to distinguish sources as fossil fuel, biomass burning and biogenic emissions on the base of $^{14}C$ measurements and levoglucosan. This revealed that summer samples exhibit an important relative contribution of non-fossil sources and a dominant contribution of biomass burning in winter. Interestingly, this very valuable data set and its important conclusions are similar to what was obtained in two source apportionment studies (Gelencsér et al., 2007; May et al., 2009) made on the basis of a two year round data set sampled on a weekly basis at five rural/remote sites in Europe. These detailed literature data set reflects atmospheric conditions of 2002/2003 on a European west east transect at altitudes from 40 to 3100 m asl. Given the fact that the source apportionment calculations were very similar than ins this study here, i.e. including also $^{14}C$ and levoglucosan measurements to distinguish fossil, biomass burning and biogenic emissions, it might be worth that the authors have a look on this dataset and benefit by comparing their new results with these existing literature data.

References:
Gelencsér, A., B. May, D. Simpson, A. Sánchez-Ochoa, A. Kasper-Giebl, H. Puxbaum, A. Caseiro, C. Pio, and M. Legrand (2007), Source apportionment of PM2.5 organic aerosol over Europe: Primary/secondary, natural/anthropogenic, and fossil/biogenic origin, J. Geophys. Res., 112, D23S04, doi:10.1029/2006JD008094.
MAY, B., WAGENBACH, D., HAMMER, S., STEIER, P., PUXBAUM, H. and PIO, C. (2009), The anthropogenic influence on carbonaceous aerosol in the European back- ground. Tellus B, 61: 464–472. doi:10.1111/j.1600-0889.2008.00379.x

We thank Dr. Preunkert for her advice about two other papers on $^{14}C$ in aerosols (Gelencser et al. 2007, May et al. 2009).

These papers published in 2007 and 2009, are based on similar apportionment calculations as in Szidat et al. (2004, 2006) already cited in our paper.

Both papers cited by Dr. Preunkert are based on the same $^{14}C$ analyses of pooled $PM_{2.5}$ aerosol samples for five European sites. Pooling aerosol filters reduces the number of $^{14}C$ analyses and allows to reach the necessary carbon amount to perform classical AMS analyses on graphite targets. Consequently, each site is only characterized by two values, one for winter and the other for summer (cf. Table 2 in Gelencsér et al. 2007, and the modified version as Table 1 in May et al. 2009).

In those papers the source apportionment is then based on the assumption of constant emission factors, e.g. $OC_{bb}$/levo and $OC_{bb}/EC_{bb}$ from the literature, notably based on test combustion in experimental fireplaces and oven.

By contrast to former works based on a few [14]C analyses, our precise study of two close sites relies on more that one hundred of [14]C analyses (duplicates of more than 50 samples), which allows to evaluate the correlation between TC, levoglucosan and [14]C in many filters even characterized by low carbon contents (thanks to the low blank and detection limit reached with the gas ion source coupled to AixMICADAS).

Based on the observed linear relationship (our Fig. 7) we were able to calculate a non-fossil carbon/levoglucosan ratio independent from the literature on test combustion. As underlined in section 3.2 of our paper, the non-fossil carbon/levoglucosan ratio derived for Passy and Chamonix is compatible with the large range reported by Schmidl et al. (2008) for test combustion on various types of wood. Our value is also compatible with the central value and range assumed by Gelencser et al. (2007) and May et al. (2009) from the literature on test combustion. As noted in our paper, our measured value based on the dual radiocarbon-levoglucosan approach agrees very well with those obtained by Zotter et al. (2014) for several Swiss stations.

As far as the apportionment calculation is concerned, the novelty of our approach (section 3.2.3) is to propose to use the ratio derived from the numerous pairs of [14]C and levoglucosan measurements, instead of relying on an assumed and uncertain emission factor.

---

## Author Response (AR2)

The authors have reasonably addressed the interactive comments and they have modified their manuscript accordingly.

However, I have several comments that need to be taken into account before this manuscript can be published in ACP.

*The authors would like to thank the editor for his careful reading. All the required modifications have been carried out and are highlighted in pink in the tracked manuscript.*

Page 2, line 23: Replace "importance in PM" by "importance of PM".

Page 2, line 28: Replace "aerosol sources" by "aerosol source".

Page 2, line 32: Replace "Different sources" by "Different source".

Page 2, line 32: Acronyms, here "CMB" and "PMF" should be defined (written full-out) when first used.

Page 3, line 1: Replace "determine contribution of the biomass" by "determine the contribution from biomass".

Page 3, line 3: Replace "demonstrated as an effective" by "demonstrated to be an effective".

Page 3, line 21: Replace "i. e." by "i.e.".

Page 4, line 16: Replace "bed supporting the" by "bed supports the".

Page 4, line 18: Replace "reduced as" by "reduced to".

Page 5, line 4: Replace "be quantify" by "be quantified".

Page 6, lines 2 and 7: It is unclear what "FC" indicates. Should it perhaps be "F14CC" instead?

Page 6, line 4: Replace "onto quartz" by "onto a quartz".

Page 6, line 13: Replace "Oxa2" by "OxA2".

Page 7, line 19: Replace "63μm" by "63 μm".

Page 7, line 23: Replace "AGE 3" by "AGE-3".

Page 7, line 29: Replace "1649a(Currie" by "1649a (Currie".

Page 8, line 14: Replace "2005))" by "2005)".

Page 9, line 9: Replace "fungis from" by "fungi from".

Page 9, line 17: Replace "samples have been analyzed" by "samples were analyzed".

Page 9, line 23: Replace "thermo-optical" by "thermal-optical".

Page 11, line 25: Replace "pointing that" by "indicating that".

Page 11, line 29: Replace "exhibit high" by "exhibits high".

Page 12, line 5: Replace "exhibit a" by "exhibits a".

Page 12, line 6: Replace "traffics over" by "traffic over".

Page 12, line 13: Replace "by (Schmidl et al., 2008)" by "by Schmidl et al. (2008)".

Page 12, line 14: Replace "agreement to those" by "agreement with those".

Page 15, line 13: Replace "have been" by "has been" and replace "emissions sources" by "emission sources".

Page 15, line 22: Replace "BVOCs emissions" by "BVOC emissions".

Page 17, line 8: Replace "suggests by" by "suggested by".

Pages 17-24, References:

1. For references with 3 or more authors, there should be a comma before the "and" preceding the last author. Note that there should be no comma before the "and" for references with only 2 authors.

2. The abbreviated journal names should contain a period (.); for example, it should be "Atmos. Chem. Phys." instead of "Atmos Chem Phys".

3. Titles of journal articles should be in lower case instead of in Title Case; thus a correction is needed for, e.g., the title in Birch and Cary (1996) and in Currie et al. (2002).

Page 24, line 21: Replace "Discuss, 12(7), 17657-17702, doi:105194/acpd-12-17657-2012" by "12, 10841-10856, doi:105194/acp-12-10841-2012"

Pages 25-30: The Table headings should be placed above the tables instead of below.

Page 25, line 6: Replace "1mgC" by "1 mgC".

Page 27, line 5: Replace "on duplicated" by "on duplicate".

Page 27, line 6: Replace "is presented" by "are presented".

Page 28, line 5: Replace "on duplicated" by "on duplicate".

Page 28, lines 5-6: Replace "is presented" by "are presented".

Page 30, line 2: Replace "Table6" by "Table 6".

Page 31, text in right ordinate of Figure 1(a): Replace "Oxa2" by "OxA2".

Page 33, line 3: Replace "measurements uncertainties" by "measurement uncertainties".

Page 36, line 5: Replace "In both case" by "In all cases".

Page 37, line 3: Replace "winter sample" by "winter samples".

Page 37, line 9: Replace "further informations" by "further information".

Page 37, line 10: Replace "originate from" by "originates from".

Page 38, line 3: Replace "winter sample" by "winter samples".

[revised manuscript text omitted]

---

## Author Response (AR3)

*The author thank the editor for his careful reading.*

*The corrections are done and ==highlighted in green== in the tracked manuscript.*

Co-Editor Decision: Reconsider after minor revisions (Editor review) (12 Oct 2016) by Willy Maenhaut

Comments to the Author:

The following corrections are still needed before this manuscript can be published in ACP.

Page 3, line 1: Replace "determine contribution of the biomass" by "determine the contribution from biomass".

Page 3, lines 22-23: Replace "graphitization, (Genberg et al., 2010, 2013) of very small samples" by "graphitization, of very small samples (Genberg et al., 2010, 2013)".

Page 15, line 13: Replace "emissions sources" by "emission sources".

Page 37, line 9: Replace "3.2.1. for further" by "3.2.1 for further".

[revised manuscript text omitted]